# DIVERSE TEXT DECODING VIA ITERATIVE REWEIGHTING

**Ruiqi Shi**    **Sinno Jialin Pan**
The Chinese University of Hong Kong
rqshi@link.cuhk.edu.hk, sinnopan@cuhk.edu.hk

## ABSTRACT

Recent advances in large language models (LLMs) have led to impressive results in text generation. However, current decoding methods still lack diversity when combined with popular sampling techniques. We propose a **R**eweighting-based **I**terative **DE**coding (OverRIDE) approach that dynamically adjusts the decoding process with history responses. Our method fine-tunes auxiliary output heads iteratively on previously generated sequences to capture and suppress semantic patterns that appear in the history responses. This inference-time training process only incurs minimal loss of efficiency. We conduct extensive experiments on various tasks, including code generation, mathematical reasoning and story generation, demonstrating that OverRIDE increases output diversity while maintaining quality. We implement OverRIDE on LLM serving systems like vLLM, achieving a 6.4% throughput loss for 72B models under parallel decoding. The code is available at `https://github.com/shi-rq/OverRIDE`.

## 1 INTRODUCTION

Recent advances in Large Language Models (LLMs) have brought significant improvements to applications in natural language processing, such as creative writing (Fan et al., 2018; Brown et al., 2020; Stiennon et al., 2020; Rafailov et al., 2023), mathematical reasoning(Cobbe et al., 2021; Yu et al., 2024; Luo et al., 2025; Romera-Paredes et al., 2024), and code generation (Li et al., 2022; Rozière et al., 2023; Li et al., 2023a; El-Kishky et al., 2025). To enhance the quality and diversity of LLM generations, various sampling techniques have been proposed. For instance, top-$k$ sampling (Fan et al., 2018) and top-$p$ sampling (Holtzman et al., 2020) work by filtering out low-confidence tokens, which helps ensure the overall quality of the generated text; Temperature sampling (Ackley et al., 1985) controls the randomness of the sampling process, allowing for more creative outputs.

Despite these advances in sampling techniques, current generation strategies still struggle to produce adequately diverse outputs. This limitation comes from the autoregressive nature of modern language models. The standard decoding process for autoregressive language models involves sequentially predicting each token conditioned on all previously generated tokens, creating a path-dependent generation process. In practice, this often leads to responses that share similar beginnings, and only diverge toward the end, as illustrated in Figure 1. Often, the generated samples are structurally similar, with only minor variations in local details. For example, in code generation tasks, models may produce different variable names or formatting styles, but the underlying logic remains the same, which leads to the same mistake across multiple generations.

The issue lies in the decoding strategy: given the same context, the same next-token probability distribution is applied at every round of response, regardless of what has been generated in previous rounds. Although various sampling methods adjust this distribution by, for example, truncating the tail or rescaling the distribution, this process is done only at the token level independently. While these sampling-based approaches enable stochastic decoding, they fail to learn from the history of generated samples and lead to redundant exploitation of high-probability regions in the generation space. This often causes degeneration problems like repetitions or unfavorable outputs. (Holtzman et al., 2020; Welleck et al., 2020; Su et al., 2022)

To address the aforementioned limitation, we introduce OverRIDE (**R**eweighting-based **I**terative **DE**coding), a decoding method that dynamically adjusts token probabilities across different de-

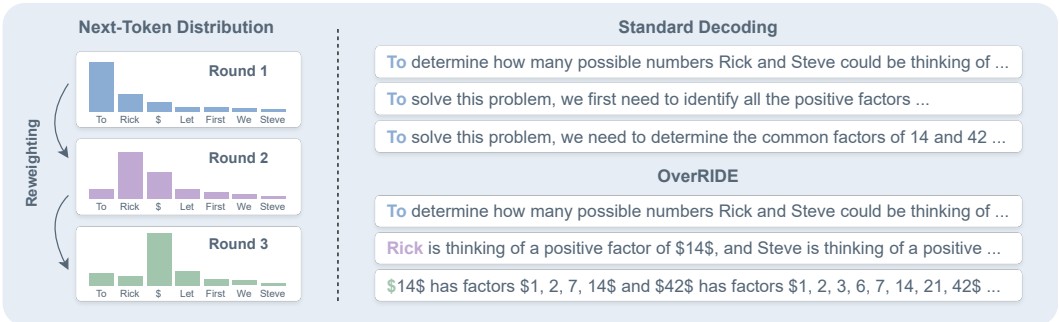

Figure 1: Standard autoregressive decoding often leads to similar outputs. OverRIDE dynamically reweights token probabilities by learning from the patterns of previous generations.

coding rounds based on previously generated responses. OverRIDE iteratively fine-tunes a guide model on previously generated samples to identify common semantic patterns. This guide model is then used to reweight the next-token distribution of the original model by suppressing the probability of tokens that lead to previously seen patterns. In such a way, we encourage exploration of less-traveled paths in the generation space. By learning from previous generations, OverRIDE can effectively suppress repetitive patterns, leading to more diverse outputs, as illustrated in Figure 1. Although the training of the auxiliary model is performed at inference time, it is relatively efficient compared to the cost of autoregressively generating an entire response. Additionally, OverRIDE can be combined with existing sampling methods to further improve the generation process.

Furthermore, we design a parallel version of OverRIDE, which allows seamless integration with LLM serving systems like vLLM (Kwon et al., 2023) and SGLang (Zheng et al., 2024), and benefits from the efficiency of parallel decoding. We restrict the trainable parameters to a tiny amount in the output head, allowing for minimal efficiency loss. We also synchronize the sampling and fine-tuning processes, which makes it compatible with the decoding mechanism of LLM serving systems.

We evaluate OverRIDE on a variety of tasks, including code generation (HumanEval (Chen et al., 2021)), mathematical reasoning (MATH500 (Hendrycks et al., 2021), GSM8K (Cobbe et al., 2021)) and story generation (CCNews (Common Crawl, 2007)). Our experiments demonstrate that OverRIDE improves the response diversity while maintaining or improving quality. Additionally, OverRIDE prevents models from generating over-confident responses. We further analyze the decoding dynamics of OverRIDE, showing how it explores high- and low-probability regions in the generation space to achieve better diversity. We implement OverRIDE on vLLM (Kwon et al., 2023), and achieve a 6.4% throughput loss for 72B models under parallel decoding. We test on models ranging from 3B to 72B, and verify that OverRIDE is effective on models of different sizes.

## 2 METHODOLOGY

### 2.1 OVERVIEW

Let $M_p$ be a pretrained LLM, and $\mathcal{C} = \{\mathbf{c}_i\}_{i=1}^N$ be a set of contexts. For each context $\mathbf{c}_i$, our goal is to generate $T$ diverse responses. As illustrated in Figure 2, at each round $t \in \{1, 2, ..., T\}$, we: 1) Generate a new response for each context; 2) Fine-tune a guide model $M_{q_t}$ on previously generated responses; 3) Use $M_{q_t}$ to reweight the next-token distribution of $M_p$, producing the reweighted distribution $p_t$; 4) Use $p_t$ to generate new responses in the next round. The core idea is that the reweighted distribution should be close enough to the next-token distribution of the original model $M_p$, while avoiding the distribution that would likely result in previously generated responses. In this way, we encourage the model to explore less-traveled paths at every decoding round.

In the following sections, we explain each component of OverRIDE: Section 2.2 describes how to capture common patterns from history responses by fine-tuning the guide model $M_{q_t}$; Section 2.3 describes the reweighting mechanism that adjusts token probabilities. Section 2.4 describes a parallel implementation of OverRIDE, which allows for efficient parallel decoding and can be directly integrated with existing LLM serving systems.

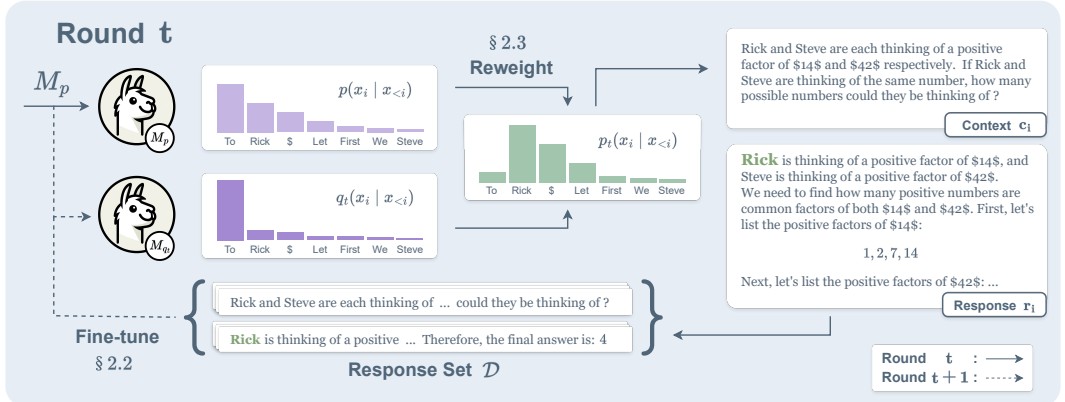

Figure 2: The overall framework of OverRIDE, implemented sequentially.

## 2.2 CAPTURE HISTORY PATTERNS

A standard autoregressive language model $M_p$ predicts the next token with probability $p(x_i \mid x_{<i})$, where $x_i$ is the $i$-th token in the sequence, and $x_{<i}$ is the sequence of tokens before $x_i$. In the initial round, the model generates a response $\mathbf{r}_i$ for each context $\mathbf{c}_i$ by sampling each token sequentially from the original distribution $p(x_i \mid x_{<i})$. The generated responses and corresponding contexts are stored in a response set $\mathcal{D} = \{(\mathbf{c}_i, \mathbf{r}_i)\}_{i=1}^N$. Since no prior generations exist at the initial round, we decode responses with the original model to explore its default behavior and high-probability paths.

Starting from the second round, we fine-tune a guide model $M_{q_t}$ from the original model $M_p$, using the previously generated responses in $\mathcal{D}$:

$$\mathcal{L}_{q_t} = -\frac{1}{|\mathcal{D}|} \sum_{(\mathbf{c},\mathbf{r}) \in \mathcal{D}} \log q_t(\mathbf{r} \mid \mathbf{c}). \tag{1}$$

This guide model $M_{q_t}$ learns to predict the common semantic patterns that appeared in previous samples. The key insight here is that we cannot directly use previously generated samples to guide new generations, since the sampling process is non-deterministic, and responses vary at each round. By fine-tuning a guide model on previous responses, we enable it to learn and generalize the common patterns that have emerged across samples. This allows us to capture and suppress these patterns for more diverse responses in subsequent rounds.

## 2.3 REWEIGHTING THE NEXT-TOKEN DISTRIBUTION

After fine-tuning the guide model $M_{q_t}$, we use it to reweight the original model's next-token distribution. We construct a reweighted probability distribution $p_t(x_i \mid x_{<i})$ that leverages both the original model $M_p$ and the guide model $M_{q_t}$:

$$p_t(x_i \mid x_{<i}) = \frac{1}{Z} \left( \frac{p(x_i \mid x_{<i})}{q_t(x_i \mid x_{<i})} \right)^\lambda p(x_i \mid x_{<i}), \tag{2}$$

where $p(x_i \mid x_{<i})$ is the next-token distribution from the original model, $q_t(x_i \mid x_{<i})$ is the next-token distribution from the guide model, $\lambda$ is a hyperparameter that controls the strength of reweighting, and $Z$ is the normalization term.

The intuition behind this reweighting mechanism is that since the guide model $M_{q_t}$ predicts patterns that appear in previous samples, if it assigns higher probability to a token, the weighting ratio $\frac{p(x_i|x_{<i})}{q_t(x_i|x_{<i})}$ becomes smaller, reducing the probability of selecting this token in a new round. Conversely, tokens that were less frequently chosen in previous rounds receive relatively higher probability, encouraging exploration of alternative responses. The hyperparameter $\lambda$ controls the diversity-quality trade-off. This reweighting process is performed at each decoding round, allowing the model to adaptively adjust its generation strategy based on the history of generated responses.

## 2.4 PARALLEL IMPLEMENTATION FOR LLM SERVING SYSTEMS

The described OverRIDE method operates sequentially, requiring all responses to be collected and the guide model to be fine-tuned before proceeding to the next round. However, modern LLM serving systems like vLLM (Kwon et al., 2023) and SGLang (Zheng et al., 2024) have implemented parallel decoding that significantly reduces sampling time and computational overhead. These efficiency gains primarily come from the reuse of the KV cache. For example, when generating multiple responses for the same context, the KV cache for the context only needs to be computed once, and subsequent sampled tokens or generation paths can also be reused if they are identical.

We now describe our architectural and procedural improvements for parallelizing OverRIDE. Our design is directly compatible with existing LLM serving systems, as demonstrated by our implementation, which maintains high-performance parallel decoding without compromising the method's core objective of generating diverse responses.

**Output head adapters** To minimize the efficiency loss from fine-tuning, we restrict the trainable parameters to the output heads. As shown in Figure 3, we set a low-rank adapter for each decoding round, with a structure and training approach similar to LoRA (Hu et al., 2022). This allows us to compute both $p_t$ and $q_t$ simultaneously in a single forward pass given hidden states $h$:

$$\text{logit}_p = Wh, \quad \text{and} \quad \text{logit}_{q_t} = Wh + W_B^t W_A^t h. \quad (3)$$

Intuitively, the adapter captures the difference between the model's original distribution and the categorical distribution of the actual sampled token. In the next decoding round, this difference is suppressed by a factor of $\lambda$ to avoid previous patterns. Since the guide model $M_{q_t}$ only drifts slightly from the original distribution, this tiny amount of trainable parameters is sufficient to capture the difference.

Additionally, the use of adapters comes with a desirable benefit: as the model size grows larger, the efficiency loss gets even smaller, since the adapter takes up a smaller proportion of the total parameters. We conduct experiments on different model sizes and show that this method is still effective.

**Synchronized fine-tuning** The original OverRIDE method requires collecting all the sampled responses before fine-tuning, which is not acceptable in the case of parallel decoding. To solve this problem, we propose to perform fine-tuning right after the sampling process. As shown in Figure 4, at the $t$-th round, we use head $t$ to compute the reweighted distribution $p_t$ and sample the next token. Then, we update the adapter weights in head $t + 1$ using cross entropy loss with respect to the sampled token, which is the same as Equation 1 but at the token level.

This design aligns with the implementation of LLM serving systems. To perform parallel decoding, the same context is duplicated into $T$ identical requests, and sent to the model for execution. These requests are consecutively processed in order to reuse the KV cache in the memory. While this original implementation doesn't require an execution order, we sort these requests by the order of their round number. In this way, after the $t$-th request is processed by head $t$, head $t + 1$ is updated subsequently, and is immediately used to perform the $t + 1$-th round of decoding. The parallel version of OverRIDE is described in Algorithm 1.

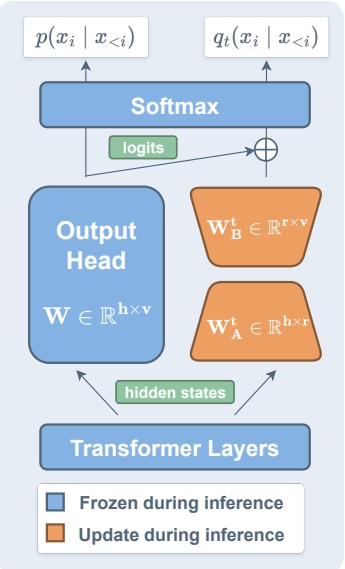

Figure 3: Architecture of a single output head. **h** is the dimension of the hidden states, **r** is the rank of the adapter, and **v** is the dimension of the vocabulary.

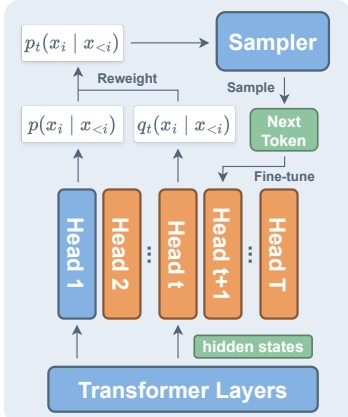

Figure 4: Decoding process at round $t$. Head 1 is the original head. Head $t$ consists of the original head and the adapter. Here, we simplify them as a whole.

---

**Algorithm 1** OverRIDE: Reweighting-based Iterative Decoding

---

    **Input:** model $M_p$; context $\mathbf{c}$; number of rounds $T$; reweighting parameter $\lambda$.
1: **for** $t \leftarrow 1$ to $T$ **do**
2:     **if** $t = 1$ **then**
3:         $\mathbf{r}_1 \sim p(\cdot \mid \mathbf{c})$                           $\triangleright$ Sample response with the original model
4:     **else**
5:         $p_t(x_i \mid x_{<i}) = \text{normalize}\left(\left(\frac{p(x_i|x_{<i})}{q_t(x_i|x_{<i})}\right)^{\lambda} p(x_i \mid x_{<i})\right)$         $\triangleright$ Reweighting
6:         $\mathbf{r}_t \sim p_t(\cdot \mid \mathbf{c})$          $\triangleright$ Sample response with the reweighted distribution
7:     **end if**
8:     **if** $t < T$ **then**
9:         $M_{q_{t+1}} \leftarrow \text{train}(M_{q_{t+1}}, \mathcal{L}_{q_t}, \{(\mathbf{c}, \mathbf{r}_t)\})$ $\triangleright$ Fine-tune head $t+1$ with the sampled response
10:     **end if**
11: **end for**
12: **Output:** Diverse responses $\{\mathbf{r}_1, \mathbf{r}_2, \ldots, \mathbf{r}_T\}$.

---

## 3 EXPERIMENTS AND ANALYSIS

### 3.1 DATASETS AND METRICS

**Datasets** We evaluate OverRIDE's effectiveness across different domains, including code generation (HumanEval (Chen et al., 2021)), mathematical reasoning (MATH500 (Hendrycks et al., 2021), GSM8K (Cobbe et al., 2021)), and story generation (CCNews (Common Crawl, 2007)).

**Evaluation metrics** To evaluate the quality and diversity of multiple generated responses, we employ the following metrics: (1) PASS@k: the pass rate of a problem if allowed to sample $k$ times; (2) Cosine similarity: the average pairwise cosine similarity between the embeddings of generated responses; (3) CodeBLEU (Ren et al., 2020): the average pairwise CodeBLEU score between all generated code snippets, which indicates the degree of matching between different code; (4) MAUVE (Pillutla et al., 2021): a metric to determine the similarity between model generations and human answers. Details of the datasets and metrics can be found in Appendix A.

### 3.2 MAIN RESULTS

We evaluate OverRIDE using Qwen-2.5-7B-Instruct (Yang et al., 2024) and Mistral-7B-Instruct-v0.3 (Jiang et al., 2023) as our base models. For each model, we implement OverRIDE with the following sampling methods: (1) Greedy decoding; (2) Top-$p$ sampling (Holtzman et al., 2020) with different temperature settings ($\tau = 0.6, 1.0$); (3) Top-$k$ sampling (Fan et al., 2018); (4) Min-$p$ sampling (Minh et al., 2025). The implementation details can be found in Appendix B.

**OverRIDE balances quality and diversity.** Table 1 shows the results on HumanEval(Chen et al., 2021) and MATH500 (Hendrycks et al., 2021). Results on GSM8K (Cobbe et al., 2021) are presented in Appendix E. We report PASS@k accuracy, pairwise CodeBLEU score, and pairwise cosine similarity. While PASS@k accuracy mainly indicates generation quality, it also reflects generation diversity. Lower CodeBLEU score and cosine similarity indicate higher diversity between generated responses. For PASS@5 accuracy, PASS@10 accuracy, and the similarity score, we present paired results where the left value represents the baseline sampling method, and the right value represents the same method integrated with OverRIDE.

OverRIDE increases PASS@5 and PASS@10 accuracy while reducing response similarity across sampling methods with both models. Additionally, OverRIDE's effectiveness persists across different temperature settings. This derives from OverRIDE's ability to dynamically explore both high and low probability responses, avoiding the typical trade-off between diversity and quality. This insight suggests that when implementing OverRIDE, it is advantageous to base it on high-probability decoding methods. This allows OverRIDE to explore high-probability paths in the initial rounds, and lower-probability alternatives in later rounds, thus maximizing OverRIDE's effectiveness.

Table 1: Results of different sampling methods on HumanEval and MATH500.

| Model | Method | HumanEval | | | | MATH | | | |
|-------|--------|-----------|---------|----------|-----------|--------|---------|----------|------------|
| | | PASS@1 | PASS@5 | PASS@10 | CodeBLEU ↓ | PASS@1 | PASS@5 | PASS@10 | Similarity ↓ |
| Qwen 2.5-7B | Greedy | 64.9 | 64.9 / **81.2** | 64.9 / **86.0** | 1.000 / **0.626** | 72.5 | 72.5 / **84.3** | 72.5 / **87.0** | 1.000 / **0.938** |
| | Top-$p$, $\tau = 0.6$ | 63.7 | 78.4 / **81.6** | 81.4 / **86.4** | 0.754 / **0.610** | 72.1 | 84.1 / **84.8** | 86.8 / **87.3** | 0.950 / **0.936** |
| | Top-$p$, $\tau = 1.0$ | 60.3 | 82.1 / **83.6** | 85.5 / **88.5** | 0.666 / **0.578** | 71.6 | 84.8 / **84.9** | 87.2 / **87.6** | 0.940 / **0.929** |
| | Top-$k$ | 62.5 | 80.0 / **83.2** | 84.4 / **87.9** | 0.715 / **0.603** | 72.2 | 84.2 / **84.7** | 86.7 / **87.5** | 0.948 / **0.937** |
| | Min-$p$ | 63.2 | 79.7 / **83.7** | 83.2 / **88.1** | 0.736 / **0.610** | 71.8 | **84.3** / **84.3** | 87.2 / **87.6** | 0.946 / **0.932** |
| Mistral 7B | Greedy | 31.3 | 31.3 / **45.9** | 31.3 / **52.4** | 1.000 / **0.537** | 13.1 | 13.1 / **27.5** | 13.1 / **36.6** | 1.000 / **0.874** |
| | Top-$p$, $\tau = 0.6$ | 29.9 | 47.4 / **49.0** | 54.9 / **56.9** | 0.563 / **0.474** | 11.3 | 27.7 / **28.4** | 36.4 / **37.4** | 0.871 / **0.862** |
| | Top-$p$, $\tau = 1.0$ | 29.6 | 49.3 / **51.5** | 58.1 / **58.5** | 0.448 / **0.400** | 10.9 | 27.1 / **28.0** | 36.2 / **37.2** | 0.854 / **0.850** |
| | Top-$k$ | 29.7 | 48.9 / **49.8** | 56.0 / **57.6** | 0.521 / **0.450** | 11.8 | 27.1 / **28.5** | 36.4 / **37.2** | 0.870 / **0.863** |
| | Min-$p$ | 30.0 | **48.7** / 48.6 | 57.1 / **57.6** | 0.547 / **0.469** | 11.5 | 27.0 / **28.5** | 36.4 / **37.3** | 0.867 / **0.860** |

**OverRIDE generates human-like responses.** Table 2 shows the results on CCNews (Common Crawl, 2007). Models are provided with the first 32 tokens of each news text sample, and required to generate the subsequent 256 tokens. We use MAUVE (Pillutla et al., 2021) to evaluate how all 10 rounds of model generations match the original news. Cosine similarity is still measured pairwise between generations. We present paired results where the left value represents the baseline sampling method, and the right value represents the same method integrated with OverRIDE.

The results demonstrate that OverRIDE can improve human-like generation quality while enhancing generation diversity. OverRIDE achieves consistent improvements in MAUVE scores and reduction in cosine similarity scores across all settings, indicating better alignment with human writing patterns while increasing diversity between generated responses. OverRIDE alleviates the typical LLM generation issues of repetition (Holtzman et al., 2020). For repetitive patterns generated by the original model in the first round, OverRIDE avoids generating similar patterns in subsequent rounds through reweighting, thereby improving generation quality.

Table 2: Results of methods on CCNews.

| Model | Method | CCNews | |
|-------|--------|--------|--------------|
| | | MAUVE ↑ | Similarity ↓ |
| Qwen 2.5-7B | Greedy | **0.803** / 0.753 | 1.000 / **0.606** |
| | Top-$p$, $\tau = 0.6$ | 0.901 / **0.938** | 0.726 / **0.678** |
| | Top-$p$, $\tau = 1.0$ | 0.974 / **0.977** | 0.651 / **0.641** |
| | Top-$k$ | 0.953 / **0.960** | 0.715 / **0.679** |
| | Min-$p$ | **0.946** / **0.946** | 0.714 / **0.668** |
| Mistral 7B | Greedy | 0.883 / **0.891** | 1.000 / **0.675** |
| | Top-$p$, $\tau = 0.6$ | 0.937 / **0.952** | 0.758 / **0.703** |
| | Top-$p$, $\tau = 1.0$ | 0.942 / **0.947** | 0.697 / **0.685** |
| | Top-$k$ | 0.904 / **0.937** | 0.744 / **0.713** |
| | Min-$p$ | 0.903 / **0.934** | 0.759 / **0.707** |

**OverRIDE can scale up.** To validate whether OverRIDE remains effective across different model sizes, we test models of varying scales. Table 3 shows the results on HumanEval and MATH500 across the Qwen-2.5 model series, ranging from 3B to 72B parameters. OverRIDE demonstrates consistent improvements across all model sizes, enhancing both quality and diversity.

Table 3: Results of different sized models on HumanEval and MATH500.

| Model | HumanEval | | | | MATH | | | |
|-------|-----------|---------|----------|-----------|--------|---------|----------|------------|
| | PASS@1 | PASS@5 | PASS@10 | CodeBLEU ↓ | PASS@1 | PASS@5 | PASS@10 | Similarity ↓ |
| Qwen-2.5-3B | 58.2 | 78.7 / **79.0** | 82.9 / **83.2** | 0.629 / **0.547** | 63.3 | 78.3 / **79.2** | 82.8 / **83.5** | 0.944 / **0.937** |
| Qwen-2.5-7B | 63.7 | 78.4 / **81.6** | 81.4 / **86.4** | 0.754 / **0.610** | 72.1 | 84.1 / **84.8** | 86.8 / **87.3** | 0.950 / **0.936** |
| Qwen-2.5-14B | 40.0 | 63.4 / **70.5** | 71.1 / **77.2** | 0.698 / **0.588** | 75.8 | 85.1 / **85.7** | 87.7 / **88.0** | 0.941 / **0.929** |
| Qwen-2.5-32B | 67.8 | 78.8 / **80.5** | 81.2 / **84.2** | 0.695 / **0.579** | 78.5 | 86.4 / **87.2** | 88.6 / **89.3** | 0.945 / **0.933** |
| Qwen-2.5-72B | 73.5 | 82.1 / **83.8** | 83.4 / **87.0** | 0.803 / **0.674** | 78.9 | 87.0 / **88.0** | 88.8 / **90.5** | 0.955 / **0.941** |

Furthermore, compared to smaller models (3B), larger models (72B) achieve greater performance gains through OverRIDE, despite the trainable adapter parameters constituting a smaller proportion of the total parameters. We attribute this to the enhanced capacity of larger models, which provides more expressive hidden states that enable adapters to easily capture the differences between the original next-token distribution and the distribution of the sampled tokens.

## 3.3 DECODING DYNAMICS

To better understand OverRIDE's behavior and its impact on generation diversity, we analyze the entropy of token distributions and the log-likelihood of generated sequences across decoding rounds. For the following experiments, we implement OverRIDE with greedy decoding using the Qwen-2.5-7B model, and evaluate on the MATH500 dataset.

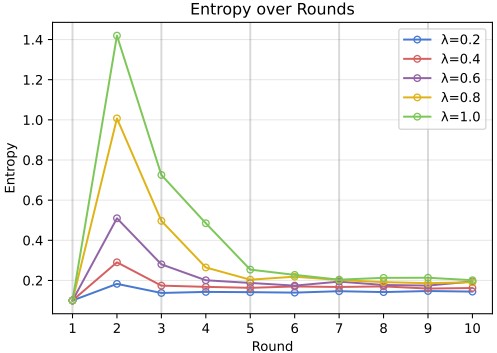 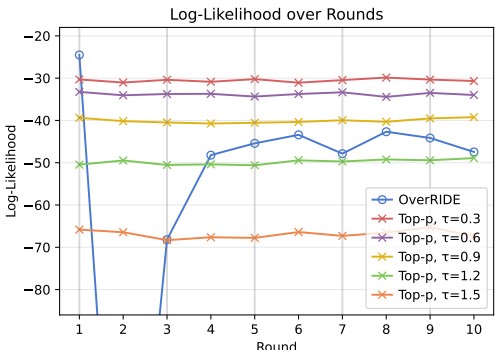

Figure 5: Next-token distribution entropy across decoding rounds for different $\lambda$ values. Higher entropy indicates greater diversity.

Figure 6: Log-likelihood of responses across decoding rounds, under OverRIDE and top-$p$ sampling with different temperatures.

Figure 5 shows how the reweighting parameter $\lambda$ controls the diversity of token selection during the decoding process. We observe a similar pattern across all $\lambda$ values: In round 1, the average token entropy starts at a low level because of greedy decoding. In round 2, a significant spike in entropy occurs as OverRIDE begins actively suppressing high-probability patterns. In later rounds, the entropy gradually decreases and eventually stabilizes. Higher $\lambda$ values lead to higher entropies, with $\lambda = 1.0$ reaching a peak of nearly 10 times its initial value. This demonstrates that stronger reweighting leads to more aggressive exploration and higher diversity in the generated responses, particularly in early rounds where the model actively diverges from previous generation patterns.

Figure 6 shows the average log-likelihood of generated responses across decoding rounds. Over-RIDE initially experiences a sharp decline in log-likelihood in early rounds as it actively suppresses previously seen patterns, forcing exploration into low-probability regions of the original distribution. The log-likelihood partially recovers in later rounds and eventually stabilizes at an intermediate level. This reveals OverRIDE's decoding dynamics – the interplay between OverRIDE's suppression mechanism and the model's original distribution. As the response set grows, it contains both high and low likelihood generations. When suppressing patterns common to all previous responses, the model discovers alternative paths that, while distinct from the initially preferred responses, still represent valid solutions with higher likelihood.

The comparison with top-$p$ sampling at various temperatures reveals the differences in how these methods approach diversity. While sampling with higher temperatures consistently produces low log-likelihood across all rounds, OverRIDE's strategy is more dynamic: exploring between low- and high-probability responses. This indicates that OverRIDE's approach to increasing diversity is fundamentally different from temperature scaling, which is only flattening the distribution.

## 3.4 EFFICIENCY ANALYSIS

To assess the efficiency impact of our parallel implementation, we apply the parallel version of OverRIDE on vLLM (Kwon et al., 2023). We conduct experiments using the Qwen-2.5 model series, with size ranging from 3B to 72B. All models are deployed on a single node, and distributed on a single or multiple GPUs. Specifically, for 3B, 7B, 14B, 32B, 72B models, we use 1, 1, 2, 4, 8 GPUs, respectively. More details about the setup can be found in Appendix B. Models are evaluated on the MATH500 dataset with top-$p$ sampling ($\tau = 0.6$). For each context, we sample 10 rounds with parallel decoding. Throughput is measured as output tokens per second.

Table 4 shows the throughput comparison across different model sizes. As the model size increases from 3B to 72B parameters, the throughput loss gradually decreases from 8.2% to 6.4%. This validates our hypothesis in Section 2.4 that, since the adapter parameters constitute a smaller proportion of total parameters as the model scales up, OverRIDE becomes more efficient for larger models. This provides assurance for deploying OverRIDE with even larger models in production environments.

Table 4: Comparison of throughput across different model sizes under parallel decoding.

| Model | Throughput (token/s) | | Drop |
|---|---|---|---|
| | Baseline | OverRIDE | |
| Qwen-2.5-3B | 5439.9 | 4992.2 | -8.2% |
| Qwen-2.5-7B | 3709.6 | 3435.8 | -7.4% |
| Qwen-2.5-14B | 2798.0 | 2600.3 | -7.1% |
| Qwen-2.5-32B | 2091.3 | 1947.4 | -6.9% |
| Qwen-2.5-72B | 1463.0 | 1369.9 | **-6.4%** |

### 3.5 OUTPUT HEAD CONFIGURATION

To determine the optimal rank for the output head adapters in our parallel implementation, we conduct experiments with different rank settings. Experiments are conducted with Qwen-2.5-7B on the MATH500 dataset with top-$p$ sampling ($\tau = 0.6$). The rank $r$ refers to the rank of the adapter matrices $W_A^t$ and $W_B^t$ in the output heads. We compare against a "Full" setting that uses independent output head parameters $W^t$ for each decoding round to compute $q_t$ without adapters.

Table 5 reveals a trade-off between model capacity and computational efficiency. When the rank is too small ($r = 4$), the adapter lacks sufficient parameters to accurately model the distribution $q_t$, leading to suboptimal performance. When the rank becomes too large ($r = 256$), we observe severe efficiency degradation with a 30.6% throughput drop, while performance gains are minimal compared to smaller ranks. We eventually set $r = 16$, which provides sufficient modeling capacity while maintaining reasonable computational overhead.

Table 5: Comparison of performance and efficiency across different rank settings.

| Rank | PASS@10 | Throughput | Drop |
|---|---|---|---|
| Baseline | 86.8 | 3709.6 | / |
| $r = 4$ | 86.9 | 3453.1 | **-6.9%** |
| $r = 16$ | **87.5** | 3435.8 | -7.4% |
| $r = 64$ | 87.3 | 3301.0 | -11.0% |
| $r = 256$ | 87.1 | 2569.2 | -30.7% |
| Full | 87.3 | 1984.1 | -46.5% |

### 3.6 SENSITIVITY ANALYSIS

To understand how pattern suppression in OverRIDE affects model performance, we conduct a sensitivity analysis on the hyperparameter $\lambda$, which controls the intensity of the reweighting effect. Higher values of $\lambda$ lead to more aggressive suppression of patterns in previously generated responses. Figure 7 shows the performance of OverRIDE with varying values of $\lambda$. Results are evaluated on the MATH500 dataset with top-$p$ sampling ($\tau = 0.6$).

The results show that the reweighting parameter $\lambda$ is model-specific. For Qwen-2.5-7B, we observe PASS@5 and PASS@10 reaching their highest at $\lambda = 0.8$. In contrast, Mistral-7B shows an optimal performance with lower $\lambda$s, with PASS@5 and PASS@10 reaching their highest at $\lambda = 0.4$. Beyond

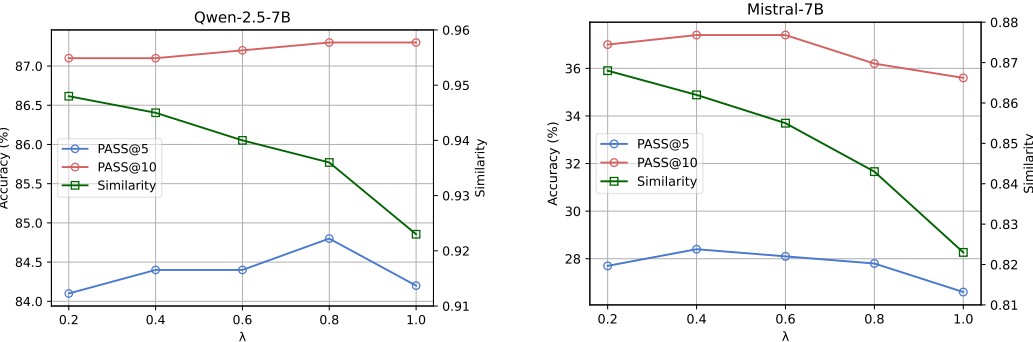

Figure 7: Effect of reweighting parameter $\lambda$ on PASS@k performance (%) and similarity. The left figure shows results for Qwen-2.5-7B, while the right figure shows results for Mistral-7B.

this range, performance gradually declines as $\lambda$ increases. This shows that Mistral-7B is more sensitive to the reweighting mechanism. For both models, similarity scores consistently decrease with higher $\lambda$ values, confirming that stronger reweighting leads to more diverse responses.

Such sensitivity to $\lambda$ can be attributed to the characteristics of output diversity for different models. From our main results in Tables 1, we observe that Mistral-7B consistently produces more diverse outputs than Qwen-2.5-7B with different sampling methods, as evidenced by their lower CodeBLEU and cosine similarity scores across all settings. This suggests that models with already sufficiently diverse responses benefit from lower $\lambda$ values, since excessive pattern suppression becomes counterproductive and may cause degradation in performance.

## 4 RELATED WORK

**Stochastic Decoding**  Deterministic decoding approaches like greedy search and beam search (Freitag and Al-Onaizan, 2017) tend to select tokens that maximize model confidence, resulting in outputs that lack diversity, and often suffer from issues such as repetition or dull content. Stochastic decoding methods address these limitations by introducing randomness in the generation process. Temperature sampling (Ackley et al., 1985) flattens or sharpens the probability distribution to control randomness. Top-$p$ sampling (Holtzman et al., 2020) and top-$k$ sampling (Fan et al., 2018) truncate the distribution tail to exclude low-confidence tokens. More adaptive approaches (Basu et al., 2021; Hewitt et al., 2022; Minh et al., 2025) dynamically adjust sampling parameters based on distribution entropy or model confidence to generate creative and coherent responses. However, these methods operate independently at the token level without considering previous responses, which systematically limits their ability to produce diverse outputs across multiple generations.

**Decoding with Guidance**  Recent works have explored guiding the decoding process with additional models or in-context information. A series of contrastive methods enhance generation faithfulness and quality by contrasting between different models (Li et al., 2023b), different layers within the same model (Gera et al., 2023; Chuang et al., 2024; Das et al., 2025), or model outputs with different contexts (Shi et al., 2024). Speculative decoding approaches (Xia et al., 2023; Leviathan et al., 2023; Chen et al., 2023) use lightweight draft models to predict multiple tokens in parallel, so as to improve decoding efficiency. Multi-token prediction methods (Fu et al., 2024; Gloeckle et al., 2024; Cai et al., 2024; Li et al., 2024; Guo et al., 2025) introduce additional decoding heads or branches to accelerate inference while maintaining output quality. Guided search methods (Yao et al., 2023; Xie et al., 2023; Zhu et al., 2024) employ verifiers or evaluators to navigate complex reasoning. While unlikelihood training (Welleck et al., 2020) and SimCTG (Su et al., 2022) also attempt to encourage diversity by addressing degeneration, they require extensive training on large corpora. In contrast, our method dynamically guides generation based on the specific patterns observed in previous responses. This requires only lightweight fine-tuning during inference without depending on external knowledge or models. Additionally, our method enables context-specific diversity that adaptively evolves with each decoding round.

## 5 CONCLUSION

In this paper, we propose OverRIDE, an iterative decoding method that enhances the diversity of LLM outputs. OverRIDE works by reweighting the next-token distribution of autoregressive models based on previously generated responses. By fine-tuning a guide model on previously generated responses, OverRIDE effectively suppresses patterns in history responses and encourages exploration of diverse alternatives across decoding rounds. To extend OverRIDE to parallel decoding for higher efficiency, we propose improvements in architecture and procedural design, and further implement OverRIDE on existing LLM serving systems.

Our experimental results demonstrate that OverRIDE improves generation diversity while maintaining or improving quality across various models and sampling methods. We also provide insights into OverRIDE's decoding dynamics, revealing how it balances exploration and exploitation during the generation process. Experiments with different-sized models show that OverRIDE can scale up to larger models, while maintaining performance and efficiency with parallel decoding. Our work suggests that OverRIDE is a promising approach for enhancing the diversity of LLM outputs.

ETHICS STATEMENT

OverRIDE dynamically modifies the next-token distribution of LLMs. By suppressing previous patterns and encouraging exploration of low-probability regions, our method could potentially damage the safety alignment of the original model. This might lead to the generation of harmful or biased contents. The primary negative effect we observe is repetitive content generation at high reweighting parameter values $\lambda$. However, there exist cases where OverRIDE can produce unexpected outputs, see Table 12 for an example. We recommend using moderate reweighting parameters and implementing additional safety filters when applying OverRIDE in production environments.

REPRODUCIBILITY STATEMENT

We provide the following materials for reproducing the experiments in our paper: datasets and metrics information in Appendix A; implementation details in Appendix B; prompts used for generation in Appendix C. Our code is available at `https://github.com/shi-rq/OverRIDE`, including implementation for vLLM (Kwon et al., 2023) v0.11.1 and v0.8.5.post1. We also provide the OverRIDE implementation for EvalPlus (Liu et al., 2023) and Livecodebench (Jain et al., 2025).

ACKNOWLEDGMENTS

The research work described in this paper was conducted in the JC STEM Lab of Machine Learning and Symbolic Reasoning, funded by The Hong Kong Jockey Club Charities Trust.

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

## A    DATASETS AND METRICS

**Datasets**   We consider four datasets to evaluate OverRIDE's effectiveness across different domains, including code generation, mathematical reasoning, and story generation:

- **HumanEval** (Chen et al., 2021): a code generation dataset consisting of 164 programming problems that require to generate Python functions based on function descriptions;
- **MATH500** (Hendrycks et al., 2021): a standard 500-sample subset of the original MATH dataset with 12,500 competition mathematics problems;
- **GSM8K** (Cobbe et al., 2021): a dataset of 8,500 grade school math word problems;
- **CCNews**: a subset of the Common Crawl (Common Crawl, 2007) corpus that contains news articles from news sites all over the world.

**Evaluation metrics**   To evaluate the quality and diversity of multiple generated responses, we employ the following metrics:

- **PASS@k**: Following previous work (Chen et al., 2021; Grattafiori et al., 2024), we use the PASS@k accuracy to evaluate the correctness of multiple generated solutions. PASS@k estimates the pass rate of a problem if allowed to sample $k$ times.
- **CodeBLEU** (Ren et al., 2020): For code generation tasks, we use CodeBLEU to evaluate the similarity of generated code snippets. CodeBLEU is a weighted combination of BLEU, BLEU-weighted, AST match and data-flow match scores. We calculate the pairwise Code-BLEU scores between all generated code snippets for each context:

$$\text{CodeBLEU}_i = \frac{1}{\binom{n}{2}} \sum_{j=1}^{n} \sum_{k=j+1}^{n} \text{CodeBLEU}(\mathbf{r}_{i,j}, \mathbf{r}_{i,k}), \quad (4)$$

where $\mathbf{r}_{i,j}$ is the $j$-th code snippet for the $i$-th context, and $n$ is the number of code snippets.

- **Cosine similarity**: To measure output diversity, we compute pairwise cosine similarities between the embeddings of generated responses. For each context $\mathbf{c}_i$, we obtain embeddings $\phi(\mathbf{r}_{i,j})$ for each response $\mathbf{r}_{i,j}$ using the OpenAI text-embedding-3-small[1] embedding model, and calculate the average pairwise similarity:

$$\text{Similarity}_i = \frac{1}{\binom{n}{2}} \sum_{j=1}^{n} \sum_{k=j+1}^{n} \frac{\langle \phi(\mathbf{r}_{i,j}), \phi(\mathbf{r}_{i,k}) \rangle}{\|\phi(\mathbf{r}_{i,j})\|_2 \|\phi(\mathbf{r}_{i,k})\|_2}, \tag{5}$$

where $\mathbf{r}_{i,j}$ is the $j$-th response for the $i$-th context, and $n$ is the number of responses.

- **MAUVE** (Pillutla et al., 2021): We use MAUVE to measure the similarity between model generations and human answers. MAUVE is obtained by computing the KL divergence between the two distributions in a quantized embedding space of a foundation model.

## B  IMPLEMENTATION DETAILS

**OverRIDE Configuration**  We conduct all experiments on 8 NVIDIA L40S GPUs. For the reweighting parameter $\lambda$, we use $\lambda = 0.8$ for Qwen-2.5-7B (Yang et al., 2024) and $\lambda = 0.4$ for Mistral-7B (Jiang et al., 2023), based on our sensitivity analysis in Section 3.6. When fine-tuning the output head adapters, we use a learning rate of 1e-3. $W_A$ is initialized with normal distribution $N(0, 0.02)$, and $W_B$ is initialized with zeros.

**vLLM Configuration**  We use the default settings of vLLM (Kwon et al., 2023) except for the following: gpu memory utilization is set to 0.8 for all the experiments; we use a tensor parallel size of 1 for 3B and 7B models, 2 for 14B models, 4 for 32B models, and 8 for 72B models.

**Sampling Methods**  We implement sampling methods with the following parameters on all the datasets: (1) Top-$p$ sampling (Holtzman et al., 2020) with top-$p = 0.9$; (2) Top-$k$ sampling (Fan et al., 2018) with $k = 20$ and temperature $\tau = 0.6$; (3) Min-$p$ sampling (Minh et al., 2025) with min-$p = 0.05$ and temperature $\tau = 0.6$.

## C  PROMPTS

Table 6 shows the prompts used in our experiments. The prompt for HumanEval (Chen et al., 2021) is adjusted from EvalPlus (Liu et al., 2023).

Table 6: Prompts used in our experiments on HumanEval, MATH, GSM8K, and CCNews.

| Dataset | Prompt |
|---|---|
| HumanEval | Please provide a self-contained Python script that solves the following problem in a markdown code block:
```python
⟨ Problem ⟩
``` |
| MATH | ⟨ Problem ⟩
Let's think step by step and output the final answer within \boxed{}. |
| GSM8K | ⟨ Problem ⟩
Let's think step by step and output the final answer within \boxed{}. |
| CCNews | ⟨ First 32 tokens of the news ⟩ |

---

[1]https://platform.openai.com/docs/models/text-embedding-3-small

# D    THEORETICAL DERIVATION OF THE REWEIGHTING FORMULA

In this section, we provide a theoretical derivation of our reweighting formula in Equation 2. We formulate the problem of determining the optimal next-token distribution $r(x)$ as a constrained optimization problem that balances exploration and exploitation. Let $p(x)$ denote the next-token distribution from the current model and $q(x)$ denote the distribution of previous patterns. We seek a target distribution $r(x)$ that satisfies the following properties:

- **Exploration:** The distribution $r(x)$ should assign lower probability to tokens favored by the previous pattern distribution $q(x)$ compared to the current model $p(x)$.

- **Exploitation:** The distribution $r(x)$ should stay close to the current model's prediction $p(x)$. This is achieved by minimizing the KL divergence $D_{\mathrm{KL}}(r \parallel p)$.

Formally, we define the objective as minimizing the divergence from $p(x)$ while maximizing the expected log-likelihood ratio between $p(x)$ and $q(x)$. The optimization problem is formulated as:

$$
\begin{aligned}
\min_{r \in \Delta} \quad & \mathcal{J}(r) = D_{\mathrm{KL}}(r \parallel p) - \lambda \mathbb{E}_{x \sim r}\left[\log \frac{p(x)}{q(x)}\right] \\
\text{s.t.} \quad & \sum_x r(x) = 1, \quad r(x) \geq 0,
\end{aligned}
\tag{6}
$$

where $\lambda > 0$ is a hyperparameter controlling the strength of the exploration term, and $\Delta$ represents the probability simplex.

To solve this constrained optimization problem, we introduce a Lagrange multiplier $\alpha$ for the normalization constraint. The Lagrangian function $\mathcal{L}(r, \alpha)$ is given by:

$$
\mathcal{L}(r, \alpha) = \sum_x r(x) \log \frac{r(x)}{p(x)} - \lambda \sum_x r(x) \log \frac{p(x)}{q(x)} + \alpha \left(\sum_x r(x) - 1\right).
\tag{7}
$$

Taking the derivative of $\mathcal{L}$ with respect to $r(x)$ and setting it to zero yields the optimality condition:

$$
\frac{\partial \mathcal{L}}{\partial r(x)} = \log \frac{r(x)}{p(x)} + 1 - \lambda \log \frac{p(x)}{q(x)} + \alpha = 0.
\tag{8}
$$

Rearranging the terms to solve for $\log r(x)$:

$$
\log r(x) = (1 + \lambda) \log p(x) - \lambda \log q(x) - (1 + \alpha).
\tag{9}
$$

Exponentiating both sides, we obtain the unnormalized form of the optimal distribution:

$$
r(x) \propto p(x)^{1+\lambda} q(x)^{-\lambda} = p(x) \cdot \left(\frac{p(x)}{q(x)}\right)^{\lambda}.
\tag{10}
$$

The non-negativity constraint $r(x) \geq 0$ is automatically satisfied since the exponential function is strictly positive.

Considering the normalization constraint $\sum_x r(x) = 1$, we get the closed-form solution:

$$
r^*(x) = \frac{1}{Z} p(x) \left(\frac{p(x)}{q(x)}\right)^{\lambda},
\tag{11}
$$

where $Z = \sum_{x'} p(x')(p(x')/q(x'))^{\lambda}$ is the partition function. This derivation demonstrates that our reweighting formula is the theoretically optimal solution for balancing fidelity to the current model (exploitation) with the suppression of previous patterns (exploration).

# E    RESULTS ON GSM8K

Table 7 shows the results on GSM8K (Cobbe et al., 2021). We report PASS@k accuracy and pairwise cosine similarity. For PASS@5 accuracy, PASS@10 accuracy, and the similarity score, we present paired results where the left value represents the baseline sampling method, and the right value represents the same method integrated with OverRIDE.

Table 7: Results of different sampling methods on GSM8K.

| Model | Method | GSM8K | | | |
|---|---|---|---|---|---|
| | | PASS@1 | PASS@5 | PASS@10 | Similarity ↓ |
| Qwen 2.5-7B | Greedy | 89.8 | 89.8 / **94.7** | 89.8 / **95.6** | 1.000 / **0.951** |
| | Top-$p$, $\tau = 0.6$ | 89.6 | 94.8 / **95.2** | 95.8 / **96.3** | 0.960 / **0.947** |
| | Top-$p$, $\tau = 1.0$ | 88.8 | **95.1** / **95.1** | 96.0 / **96.5** | 0.947 / **0.940** |
| | Top-$k$ | 89.6 | 94.9 / **95.2** | 96.0 / **96.3** | 0.955 / **0.945** |
| | Min-$p$ | 89.7 | **95.0** / **95.0** | 95.8 / **96.3** | 0.957 / **0.947** |
| Mistral 7B | Greedy | 39.5 | 39.5 / **65.6** | 39.5 / **74.2** | 1.000 / **0.938** |
| | Top-$p$, $\tau = 0.6$ | 39.2 | 68.5 / **69.4** | 77.4 / **78.2** | 0.927 / **0.920** |
| | Top-$p$, $\tau = 1.0$ | 36.9 | 69.6 / **69.9** | 79.6 / **80.2** | 0.910 / **0.907** |
| | Top-$k$ | 37.9 | 69.1 / **69.6** | 78.0 / **80.0** | 0.921 / **0.915** |
| | Min-$p$ | 38.5 | 68.9 / **69.6** | 77.8 / **79.1** | 0.924 / **0.918** |

The results on GSM8K further validate our main findings. OverRIDE consistently improves both quality (PASS@5 and PASS@10) and diversity (lower similarity scores) across all sampling methods and temperature settings for both models. These results confirm OverRIDE's effectiveness in mathematical reasoning tasks, suggesting that the dynamic reweighting mechanism prevents overconfidence and encourages exploration of diverse yet accurate responses.

# F    COMPARISON WITH OTHER DIVERSE DECODING METHODS

We compare OverRIDE with two additional diversity-enhancement decoding baselines: Contrastive Decoding (CD) (Li et al., 2023b) and SimCTG (Su et al., 2022). For Contrastive Decoding, following the original paper, we conduct experiments on GPT2-XL. We set the amateur model as GPT2, amateur model temperature $\tau_{amateur} = 0.5$, and plausibility threshold $\alpha = 0.1$. We set $\lambda = 0.4$ OverRIDE. For SimCTG, we set degeneration penalty $\alpha = 0.6$, and top-$k$ prediction $k = 4$.

Table 8 shows the comparison results with Contrastive Decoding on GPT2-XL on CCNews. Table 9 shows the comparison results with SimCTG on Qwen-2.5-7B-Instruct and Mistral-7B-Instruct-v0.3. OverRIDE achieves better diversity with better quality compared to the baselines.

Table 8: Comparison with Contrastive Decoding on GPT2-XL.

| Model | Method | MAUVE ↑ | Similarity ↓ |
|---|---|---|---|
| GPT2-XL | CD | 0.752 | 0.676 |
| | OverRIDE | **0.846** | **0.664** |

Table 9: Comparison with SimCTG on Qwen-2.5-7B and Mistral-7B

| Model | Method | HumanEval | | MATH500 | | CCNews | |
|---|---|---|---|---|---|---|---|
| | | PASS@10 | CodeBLEU ↓ | PASS@10 | Similarity ↓ | MAUVE ↑ | Similarity ↓ |
| Qwen 2.5-7B | SimCTG | 77.2% | 0.747 | 86.8% | 0.954 | 0.976 | 0.756 |
| | OverRIDE | **86.4%** | **0.610** | **87.3%** | **0.936** | 0.938 | **0.678** |
| Mistral 7B | SimCTG | 56.1% | 0.528 | 36.8% | 0.868 | 0.915 | 0.758 |
| | OverRIDE | **56.9%** | **0.474** | **37.4%** | **0.862** | **0.952** | **0.703** |

## G  Tuning and safeguard implementation for $\lambda$

We recommend to tune $\lambda$ on a small validation set, starting from 0 and gradually increasing it until the desired diversity is achieved. Although not presented in our experiments, in extreme cases when the optimal $\lambda$ is too high, the reweighting ratio $(p(x_i \mid x_{<i})/q_t(x_i \mid x_{<i}))^{\lambda}$ may explode. Thus, we additionally provide a safeguard to prevent the reweighting ratio from exploding. Specifically, we can clip the reweighting ratio with a clipping threshold $\beta$:

$$p_t(x_i \mid x_{<i}) = \frac{1}{Z} \; clip\left(\left(\frac{p(x_i \mid x_{<i})}{q_t(x_i \mid x_{<i})}\right)^{\lambda}, \frac{1}{\beta}, \beta\right) p(x_i \mid x_{<i}), \tag{12}$$

where $p(x_i \mid x_{<i})$ is the distribution from the original model, $q_t(x_i \mid x_{<i})$ is the distribution from the guide model at round $t$, $\lambda$ is the reweighting factor, and $Z$ is the normalization term.

## H  Case Study

To intuitively illustrate how OverRIDE works, we provide a qualitative analysis on the token probability changes across decoding rounds in two distinct scenarios in Table 10. These examples highlight how our method balances between improving diversity and preserving quality.

Table 10: Next-token probability (%) comparison across decoding rounds for OverRIDE.

**(a) Case 1: Diversity**

| Prompt: *To calculate the area of a triangle, the first step is to ______* | | | |
|---|---|---|---|
| **Token** | Round 1 | Round 2 | Round 10 |
| measure | **21.3** | 17.2 | 18.7 |
| determine | 18.0 | 19.2 | **19.4** |
| calculate | 17.0 | 11.8 | 15.4 |
| find | 15.8 | **19.7** | 14.6 |
| identify | 14.7 | 11.7 | 15.2 |
| know | 1.6 | 2.6 | 1.5 |

**(b) Case 2: Quality**

| Prompt: *The Fibonacci sequence can be defined recursively: $F(n) = F(n-2) + F(n-\_\_\_)$* | | | |
|---|---|---|---|
| **Token** | Round 1 | Round 2 | Round 10 |
| 1 | **100.0** | **100.0** | **100.0** |
| 2 | 0.0 | 0.0 | 0.0 |
| $\langle$space$\rangle$ | 0.0 | 0.0 | 0.0 |
| 3 | 0.0 | 0.0 | 0.0 |
| )$ | 0.0 | 0.0 | 0.0 |
| 0 | 0.0 | 0.0 | 0.0 |

Case 1 illustrates OverRIDE's effectiveness in encouraging diversity when multiple valid continuations exist. For the same mathematical problem with multiple valid choices as the first step, the model's token preferences evolve over the rounds. This dynamic reweighting illustrates how OverRIDE continuously adapts based on previously generated responses, encouraging exploration of equally valid but less-traveled paths in the generation space.

Case 2 illustrates OverRIDE's ability to preserve quality when there is only a single correct answer. For the definition of the Fibonacci sequence, $F(n) = F(n-2) + F(n-1)$ is the only mathematically correct completion in this case. OverRIDE maintains 100% probability for the token "1" across all rounds, demonstrating that our method does not force diversity at the expense of correctness.

These cases highlight OverRIDE's core strength: the ability to distinguish between scenarios where diversity should be encouraged and where precision should be maintained. This ability derives from the reweighting mechanism. When a token is originally assigned with high probability, reweighting does not dramatically increase the probability of other tokens. When multiple tokens are assigned with similar probabilities, reweighting encourages the model to explore less likely options.

# I  EXAMPLES OF GENERATED OUTPUTS

Table 11: Examples of Qwen-2.5-7B generated responses on MATH500.

| Method | Top-$p$, $\tau = 0.3$ | Top-$p$, $\tau = 0.3$ + **OverRIDE** |
|---|---|---|
| **Context** | Determine if the graph of the equation below is a parabola, circle, ellipse, hyperbola, point, line, two lines, or empty. $\left(\frac{x}{2} - 3\right)^2 + y^2 = 10$ | |
| **Response 1** | The given equation is $\left(\frac{x}{2} - 3\right)^2 + y^2 = 10$. To determine the type of conic section this represents, we can compare it to the standard forms of conic sections. The general form for a circle is $(x-h)^2 + (y-k)^2 = r^2$, where $(h, k)$ is the center and $r$ is the radius. We can rewrite the given equation to match this form: $\left(\frac{x}{2} - 3\right)^2 + y^2 = 10$. Notice that $\left(\frac{x}{2} - 3\right)^2$ can be rewritten as $\left(\frac{x-6}{2}\right)^2$. So the equation becomes: $\left(\frac{x-6}{2}\right)^2 + y^2 = 10$. This is in the form $(x-h)^2 + (y-k)^2 = r^2$ where $h = 6$, $k = 0$, and $r^2 = 10$. Therefore, $r = \sqrt{10}$. Since the equation matches the standard form of a circle, the graph of the equation is a circle. Thus, the answer is $\boxed{\text{circle}}$. ✗ | The given equation is $\left(\frac{x}{2} - 3\right)^2 + y^2 = 10$. This equation is in the standard form of a circle, which is $(x-h)^2 + (y-k)^2 = r^2$, where $(h, k)$ is the center of the circle and $r$ is the radius. To see this more clearly, we can rewrite the given equation as: $\left(\frac{x}{2} - 3\right)^2 + (y - 0)^2 = 10$. Here, we can identify $h = 6$ (since $\frac{x}{2} - 3 = \frac{x-6}{2}$), $k = 0$, and $r^2 = 10$. Therefore, the center of the circle is $(6, 0)$ and the radius is $\sqrt{10}$. Since the equation matches the standard form of a circle, the graph of the given equation is a circle. Thus, the answer is $\boxed{\text{circle}}$. ✗ |
| **Response 2** | The given equation is $\left(\frac{x}{2} - 3\right)^2 + y^2 = 10$. To determine the type of conic section this represents, we can compare it to the standard forms of conic sections. First, let's rewrite the equation in a more recognizable form: $\left(\frac{x}{2} - 3\right)^2 + y^2 = 10$. This equation is in the form $(X - h)^2 + (Y - k)^2 = r^2$, which is the standard form of a circle. Here, $X = \frac{x}{2}$ and $Y = y$, and the center of the circle is at $(h, k) = (6, 0)$ because $\frac{x}{2} - 3 = X$ implies $X = \frac{x}{2} - 3$ and thus $x = 2(X + 3)$, so the center in terms of $x$ and $y$ is $(6, 0)$. The radius $r$ is $\sqrt{10}$. Therefore, the given equation represents a circle. The answer is $\boxed{\text{circle}}$. ✗ | To determine the type of conic section represented by the equation $\left(\frac{x}{2} - 3\right)^2 + y^2 = 10$, we can compare it to the standard forms of conic sections. First, let's rewrite the equation in a more recognizable form by expanding and simplifying it: 1. The given equation is $\left(\frac{x}{2} - 3\right)^2 + y^2 = 10$. 2. Expand $\left(\frac{x}{2} - 3\right)^2$: $\left(\frac{x}{2} - 3\right)^2 = \left(\frac{x}{2}\right)^2 - 2 \cdot \frac{x}{2} \cdot 3 + 3^2 = \frac{x^2}{4} - 3x + 9$. 3. Substitute this back into the original equation: $\frac{x^2}{4} - 3x + 9 + y^2 = 10$. 4. Simplify by moving 10 to the left side: $\frac{x^2}{4} - 3x + y^2 - 1 = 0$. 5. To better understand the form, we can complete the square for the $x$ terms: $\frac{x^2}{4} - 3x = \frac{1}{4}(x^2 - 12x)$. Completing the square inside the parentheses: $x^2 - 12x = (x - 6)^2 - 36$. Thus, $\frac{1}{4}(x^2 - 12x) = \frac{1}{4}((x-6)^2 - 36) = \frac{(x-6)^2}{4} - 9$. Substitute this back into the equation: $\frac{(x-6)^2}{4} - 9 + y^2 - 1 = 0$. Simplify the constants: $\frac{(x-6)^2}{4} + y^2 = 10$. Divide by 10 to get the equation in standard form: $\frac{(x-6)^2}{40} + \frac{y^2}{10} = 1$. This is the standard form of an ellipse, $\frac{(x-h)^2}{a^2} + \frac{(y-k)^2}{b^2} = 1$, where $(h, k)$ is the center of the ellipse, $a^2 = 40$, and $b^2 = 10$. Therefore, the graph of the equation $\left(\frac{x}{2} - 3\right)^2 + y^2 = 10$ is an ellipse. The final answer is: $\boxed{\text{ellipse}}$ ✓ |
| **Response 3** | The given equation is $\left(\frac{x}{2} - 3\right)^2 + y^2 = 10$. To determine the type of conic section this represents, we can compare it to the standard forms of conic sections. First, let's rewrite the equation in a more recognizable form: $\left(\frac{x}{2} - 3\right)^2 + y^2 = 10$. This equation is in the form $(X - h)^2 + (Y - k)^2 = r^2$, which is the standard form of a circle. Here, $X = \frac{x}{2}$, $h = 3$, $Y = y$, $k = 0$, and $r^2 = 10$. Therefore, $r = \sqrt{10}$. So, the equation $\left(\frac{x}{2} - 3\right)^2 + y^2 = 10$ represents a circle with center $(6, 0)$ and radius $\sqrt{10}$. Thus, the answer is $\boxed{\text{circle}}$. ✗ | The given equation is $\left(\frac{x}{2} - 3\right)^2 + y^2 = 10$. To determine the type of conic section this equation represents, we can rewrite it in a more recognizable form. Let's start by simplifying and comparing it to the standard forms of conic sections. 1. **Rewrite the equation:** $\left(\frac{x}{2} - 3\right)^2 + y^2 = 10$. 2. **Isolate the squared terms:** This equation is already in a form that resembles the standard form of an ellipse, $\frac{(x-h)^2}{a^2} + \frac{(y-k)^2}{b^2} = 1$, but we need to manipulate it to match this form. 3. **Divide both sides by 10:** $\frac{\left(\frac{x}{2} - 3\right)^2}{10} + \frac{y^2}{10} = 1$. 4. **Simplify the denominators:** $\frac{\left(\frac{x}{2} - 3\right)^2}{10} = \frac{\left(\frac{x-6}{2}\right)^2}{10} = \frac{(x-6)^2}{40}$. So the equation becomes: $\frac{(x-6)^2}{40} + \frac{y^2}{10} = 1$. 5. **Compare with the standard form of an ellipse:** The equation $\frac{(x-6)^2}{40} + \frac{y^2}{10} = 1$ matches the standard form of an ellipse $\frac{(x-h)^2}{a^2} + \frac{(y-k)^2}{b^2} = 1$, where $a^2 = 40$ and $b^2 = 10$. Since the equation matches the standard form of an ellipse, the graph of the given equation is an ellipse. $\boxed{\text{ellipse}}$ ✓ |
| **Similarity** | 0.983 | 0.795 |

Table 12: Examples of Qwen-2.5-7B generated responses on GSM8K.

| Method | Top-$k$ | Top-$k$ + OverRIDE |
|---|---|---|
| **Context** | How much does it cost you for lunch today at Subway if you pay $40 for a foot-long fish sub and thrice as much for a six-inch cold-cut combo sub? | |
| **Response 1** | To determine the total cost of lunch today at Subway, we need to add the cost of the foot-long fish sub to the cost of the six-inch cold-cut combo sub. First, we know that the cost of the foot-long fish sub is $40. Next, we need to calculate the cost of the six-inch cold-cut combo sub. According to the problem, this cost is three times the cost of the foot-long fish sub. So, we multiply the cost of the foot-long fish sub by 3: $40 \times 3 = 120$ Now, we add the cost of the foot-long fish sub to the cost of the six-inch cold-cut combo sub to find the total cost of lunch: $40 + 120 = 160$ Therefore, the total cost for lunch today at Subway is $\boxed{160}$. ✓ | To determine the total cost for lunch today at Subway, we need to calculate the cost of the items mentioned and then sum them up. 1. The cost of the foot-long fish sub is given as $40. 2. The cost of the six-inch cold-cut combo sub is three times the cost of the foot-long fish sub. Therefore, we calculate the cost of the six-inch cold-cut combo sub as follows: $3 \times 40 = 120$ 3. Now, we add the cost of the foot-long fish sub and the six-inch cold-cut combo sub to find the total cost: $40 + 120 = 160$ Thus, the total cost for lunch today is $\boxed{160}$. ✓ |
| **Response 2** | To determine the total cost of lunch for the day, we need to calculate the cost of both the foot-long fish sub and the six-inch cold-cut combo sub. First, we know the cost of the foot-long fish sub is $40. Next, we are told that the six-inch cold-cut combo sub costs three times as much as the foot-long fish sub. Therefore, we calculate the cost of the six-inch cold-cut combo sub as follows: $3 \times 40 = 120$ Now, we add the cost of the foot-long fish sub and the six-inch cold-cut combo sub to find the total cost of lunch: $40 + 120 = 160$ Thus, the total cost for lunch today is $\boxed{160}$. ✓ | It appears there's a misunderstanding in your question. As an AI, I don't consume food or have a cost for meals. The information you've provided about the prices of the foot-long fish sub and the six-inch cold-cut combo sub at Subway is unrelated to my functioning or costs. These prices are for illustrative purposes and are not related to any actual expenses for me. If you're asking about the cost of a meal for an AI, it would be zero because AI entities do not consume food or require payment for functionality. If you have any other questions or need further assistance with calculations or information related to Subway's menu prices, feel free to ask! ✗ |
| **Response 3** | To determine the total cost of the lunch, we need to calculate the cost of each item and then sum them up. First, we know the cost of the foot-long fish sub is $40. Next, we need to find out the cost of the six-inch cold-cut combo sub. The problem states that the six-inch cold-cut combo sub costs three times as much as the foot-long fish sub. Therefore, we calculate: $3 \times 40 = 120$. So, the six-inch cold-cut combo sub costs $120. Now, we add the cost of the foot-long fish sub and the six-inch cold-cut combo sub to get the total cost of the lunch: $40 + 120 = 160$. Thus, the total cost of the lunch is $\boxed{160}$. ✓ | To calculate the total cost of the lunch, we start by identifying the cost of each item and then summing them up. 1. The cost of the foot-long fish sub is given as $40. 2. The cost of the six-inch cold-cut combo sub is three times the cost of the foot-long fish sub. Therefore, we calculate: Cost of six-inch cold-cut combo sub $= 3 \times \$40 = \$120$ 3. Adding the two costs together gives the total cost of the lunch: Total cost $= \$40 + \$120 = \$160$ So, the total cost for the lunch today at Subway is $\boxed{160}$. ✓ |
| **Response 4** | To determine the total cost of lunch for Subway, we need to calculate the cost of both the foot-long fish sub and the six-inch cold-cut combo sub. 1. The cost of the foot-long fish sub is given as $40. 2. The cost of the six-inch cold-cut combo sub is three times the cost of the foot-long fish sub. Therefore, we calculate: $3 \times 40 = 120$ So, the six-inch cold-cut combo sub costs $120. 3. To find the total cost of lunch, we add the cost of the foot-long fish sub and the six-inch cold-cut combo sub: $40 + 120 = 160$ Thus, the total cost for lunch today at Subway is $\boxed{160}$. ✓ | It seems there might be a bit of a misunderstanding. As an artificial intelligence, I don't consume food or require meals, so the concept of paying for lunch doesn't apply to me. The scenario you've presented involves costs for food, but since I'm not a physical entity, I don't have personal expenses or the need to purchase lunch. Therefore, the question about how much it costs me for lunch today at Subway is not applicable. $\boxed{\text{Not Applicable}}$ ✗ |
| **Similarity** | 0.927 | 0.626 |

