# OpenReview forum: "Diverse Text Decoding via Iterative Reweighting"
_ICLR.cc/2026/Conference — ICLR 2026 Poster_

### Official Review · Reviewer_BHNo · 2025-10-30

**Soundness:** 3
**Presentation:** 3
**Contribution:** 3
**Rating:** 6
**Confidence:** 3

**Summary:**

This paper introduces OverRIDE (Reweighting-based Iterative DEcoding), a novel decoding method that enhances diversity in LLM text generation by learning from previously generated responses. The key innovation is fine-tuning a guide model on past generations and using it to reweight the next-token distribution, thereby suppressing repetitive patterns. The authors implement a parallel version compatible with vLLM, achieving minimal throughput loss. Experiments on code generation, mathematical reasoning, and story generation demonstrate improved diversity while maintaining or improving quality.

**Strengths:**

1. Learning from generation history is an elegant solution to the diversity problem that goes beyond token-level adjustments.
2. Strong experimental validation across multiple domains (code, math, story), model sizes (3B-72B), and sampling methods.
3. The parallel version with vLLM integration shows the authors considered real-world deployment.
4. The decoding dynamics analysis (Section 3.3) provides valuable insights into how the method works, showing the interplay between exploration and quality.

**Weaknesses:**

1. No comparison with other diversity-enhancement methods like SimCTG, contrastive decoding variants, or other inference-time techniques. The paper only compares against standard sampling.
2. Why is the reweighting formula in Equation 2 the right choice? No analysis of convergence properties or theoretical guarantees. Connection to exploration-exploitation tradeoffs not formalized.

**Questions:**

1. How does OverRIDE compare to other diversity methods like SimCTG or contrastive decoding? A direct comparison would strengthen the claims.
2. Can you provide theoretical analysis or intuition for why the specific reweighting formula works better than alternatives (e.g., additive vs. multiplicative)?
3. How do you recommend selecting λ for a new model or task without extensive hyperparameter search?
4. What is the computational breakdown between inference and adapter fine-tuning? How many gradient steps per round?

---

> ### Author Response · Authors · 2025-11-23
> **Rebuttal from authors (1/2)**
>
> Thank you for your valuable comments. We hope the following responses can address your concerns.
>
> > **Q1. Comparison with other diversity-enhancement methods.**
>
> Below, we compare OverRIDE with Contrastive Decoding [1] and SimCTG [2]. Here are the settings for these baselines:
>
> - Contrastive Decoding (CD): We use the implementation from their official Github repo. Since they only support models before 2023, and haven't responded to requests for adding recent models, we are only able to compare with OverRIDE on `GPT2-XL`. Following the original implementation, we set the amateur model as `GPT2`, amateur model temperature $\tau_{amateur}=0.5$, and plausibility threshold $\alpha=0.1$. For OverRIDE, we set $\lambda=0.4$.
>
> - SimCTG: We use the implementation integrated in Huggingface transformers. We set degeneration penalty $\alpha=0.6$, and top-$k$ prediction $k=4$.
>
> Contrastive Decoding:
>
> | Model | Method | MAUVE (CCNews) | Similarity (CCNews) |
> |-------|-------|-------|-------|
> | GPT2-XL | CD | 0.752 | 0.676 |
> | | OverRIDE | **0.846** | **0.664** |
>
> SimCTG:
>
> | Model | Method | PASS@10 (HumanEval) | CodeBLEU (HumanEval) | PASS@10 (MATH500) | Similarity (MATH500) | MAUVE (CCNews) | Similarity (CCNews) |
> |-------|-------|-------|-------|-------|-------|-------|-------|
> | Qwen-2.5-7B-Instruct | SimCTG | 77.2% | 0.747 | 86.8% | 0.954 | **0.976** | 0.756 |
> | | OverRIDE | **86.4%** | **0.610** | **87.3%** | **0.936** | 0.938 | **0.678** |
> | Mistral-7B-Instruct-v0.3 | SimCTG | 56.1% | 0.528 | 36.8% | 0.868 | 0.915 | 0.758 |
> | | OverRIDE | **56.9%** | **0.474** | **37.4%** | **0.862** | **0.952** | **0.703** |
>
>
> > **Q2. Why is the proposed reweighting formula the right choice?**
>
> We can show you how our formula is a simple yet effective way to achieve our motivation. When we design the reweighting formula, we consider the following two factors:
>
> 1. Previously generated patterns should be suppressed.
>
> 2. The reweighted distribution should be close enough to the original model's next-token distribution, otherwise it will generate broken responses.
>
> To achieve the first goal, we use $(p(x) / q(x))^\lambda$ as a reweighting factor close to 1. Since $q(x)$ captures the previously generated patterns, this reweighting factor will suppress the previously generated patterns. To achieve the second goal, we apply the factor to the original distribution $p(x)$. These designs result in the final formula: $p_t(x) = \frac{1}{Z}  (p(x) / q(x))^\lambda  p(x)$.
>
> Other simpler choices, as you have mentioned, additive: $p_t(x) = p(x) - \lambda q(x)$ or multiplicative: $p_t(x) = (p(x) / q(x))^\lambda$, mainly violate the second goal. They may cause the distribution to deviate drastically from the original distribution, leading to broken responses, and making it difficult to determine the valid hyperparameters.
>
> In case you find this intuition still not convincing enough, below we provide a non-rigorous proof by formulating our motivation as a constrained optimization problem:
>
> ---
>
> We want to find an optimal next-token distribution $r(x)$ that:
>
> - stay close to $p(x)$ (exploitation).
>
> - have low probability on previous patterns $q(x)$ comparing to $p(x)$ (exploration).
>
> Thus, we want to solve the following optimization problem:
>
> $$
> \min_{r \in \Delta} ~~ D_{KL}(r \mid p) - \lambda \mathbb{E}_{x\sim r} \left[ \log \frac{p(x)}{q(x)} \right],
> $$
>
> $$
> s.t. ~~ \sum_{x} r(x) = 1, ~~ r(x) \geq 0,
> $$
>
> where $\Delta = \{r : \sum_{x} r(x) = 1, ~~ r(x) \geq 0\}$.
>
> We can introduce a Lagrange multiplier $\alpha$ to handle the constraint:
>
> $$
> \mathcal{L}(\alpha, r) = D_{KL}(r \mid p) - \lambda \mathbb{E}_{x\sim r} \left[ \log \frac{p(x)}{q(x)} \right] + \alpha (\sum_x r(x) - 1),
> $$
>
> Taking the derivative of $\mathcal{L}(\alpha, r)$ with respect to $r$ and setting it to 0, we get:
>
> $$
> \log r(x) = (1 + \lambda) \log p(x) - \lambda \log q(x) - 1 - \alpha.
> $$
>
> Taking the exponential of both sides, and considering $\sum_{x} r(x) = 1$, we get:
>
> $$
> r(x) = \frac{1}{Z}  \left( \frac{p(x)}{q(x)} \right)^\lambda  p(x),
> $$
>
> where $Z = \sum_{x} (p(x) / q(x))^\lambda  p(x)$.
>
> This is exactly the same as our reweighting formula.
>
> ---
>
> [1] Li X L, Holtzman A, Fried D, et al. Contrastive decoding: Open-ended text generation as optimization[C]//Proceedings of the 61st annual meeting of the association for computational linguistics (volume 1: Long papers). 2023: 12286-12312.
>
> [2] Su Y, Lan T, Wang Y, et al. A contrastive framework for neural text generation[J]. Advances in Neural Information Processing Systems, 2022, 35: 21548-21561.

---

> ### Author Response · Authors · 2025-11-23
> **Rebuttal from authors (2/2)**
>
> > **Q3. How do you recommend selecting λ for a new model or task without extensive hyperparameter search?**
>
> The hyperparameter $\lambda$ is actually extremely easy and efficient to tune. Since $\lambda$ controls the intensity of the reweighting ratio $(p(x) / q(x))^\lambda$, when setting to $\lambda=0$, OverRIDE is the same as the original model's next-token distribution. Thus, we recommend to tune $\lambda$ starting from 0 and gradually increasing it until the desired diversity is achieved. For less diverse models like Qwen, we have shown in our experiment that the optimal $\lambda$ is around 0.8. For more diverse models like Mistral and Llama, the optimal $\lambda$ is even lower, thus relieves the effort of hyperparameter tuning.
>
> The best practice is to tune $\lambda$ on a small validation set. When using OverRIDE on a new model or task, we recommend to take around 100 samples and tune $\lambda$ starting from 0. Although the hardware configuration / memory settings / model / task may vary, with the help of our parallel implementation on LLM serving systems like vLLM, tuning one round typically takes **minutes**. Furthermore, reaching the optimal $\lambda$ usually takes only a few rounds.
>
> In case you still have concerns about the tuning process when $\lambda$ is too high, we additionally provide a safeguard to prevent the reweighting ratio from exploding. Specifically, users can choose to clip the reweighting ratio $(p(x) / q(x))^\lambda$ with a clipping threshold $\beta$:
>
> $$
> p_t(x) = \frac{1}{Z} ~~ clip \left( \left( \frac{p(x)}{q(x)} \right)^\lambda, ~\frac{1}{\beta}, ~\beta \right)  p(x),
> $$
>
> We have experimented with this safeguard on large $\lambda=1.5$, and show that our safeguard is effective in preventing the reweighting ratio from exploding, and helps achieve better decoding quality. Experiments are conducted on HumanEval using top-$p$ sampling with top-$p=0.9$ and $\tau=0.6$:
>
> | Model | $\beta$ | PASS@10 |
> |-------|-------|-------|
> | Qwen-2.5-7B-Instruct | No clip | 85.7% |
> | | 1.1 | 86.0% |
> | | 1.5 | **86.4%** |
> | | 2.0 | 86.1% |
> | Mistral-7B-Instruct-v0.3 | No clip | 36.0% |
> | | 1.1 | 34.2% |
> | | 1.5 | **39.0%** |
> | | 2.0 | **39.0%** |
>
> Note that this is only to provide an additional option for users to tune on extreme cases (e.g. when a much less diverse model is used). For moderate $\lambda$ values, the reweighting ratio is unlikely to explode, and the original tuning process is sufficient. Usually the optimal $\lambda$ falls into the normal situation.
>
>
>
> > **Q4. What is the computational breakdown between inference and adapter fine-tuning? How many gradient steps per round?**
>
> Each time a continuous batch of requests is processed by the LLM engine, corresponding adapters are updated for **1** gradient step. This process is closely related with the mechanism of the serving system. Allow me to walk you through the process with vLLM as an example:
>
> An important concept to notice is that multiple requests are symultanously processed by the LLM engine to calculate the next token. In the case of parallel decoding, say we plan to process 5 prompts with 3 rounds each, 15 requests are added to the running list. If we denote $R_i^t$ as the i-th request at the t-th round, we get a list of requests:
>
> > $R_1^1$, $R_1^2$, $R_1^3$, $R_2^1$, $R_2^2$, $R_2^3$, $R_3^1$, $R_3^2$, $R_3^3$, $R_4^1$, $R_4^2$, $R_4^3$, $R_5^1$, $R_5^2$, $R_5^3$.
>
> Based on the allocated memory for KV cache, the scheduler estimates how many requests can be processed as a continuous batch. These requests may be at different progress of decoding, and requests may be added (memory available) or removed (decoding finished) from the batch dynamically. Take an example in one of our Qwen-2.5-7B experiments, 19.77 GiB of memory is allocated for KV cache, and 370096 tokens are estimated to be processed by the model, thus the maximum concurrent requests is 45.18 for 8192 tokens per request.
>
> Once the batch is determined, the LLM engine will process all the requests in the batch concurrently. Hidden states are calculated, and then the next token is sampled for each request. It is at this moment that the corresponding adapters are updated for 1 gradient step, based on which decoding round $t$ the request is at.
>
> Hopefully, we have clearly explained the process. We're more than happy to answer any further questions.

---

> ### Author Response · Authors · 2025-11-26
> **Thank you for your efforts in improving OverRIDE!**
>
> Dear reviewer BHNo,
>
> Thanks to your suggestions on extensive experiments, we are able to further validate that OverRIDE is indeed an effective and robust decoding method.
>
> If you have further questions and doubts, we are always glad to provide more clarification.
>
> Otherwise, if you find our work more solid after we respond, could you consider raising your score to the corresponding level?

---

### Official Review · Reviewer_Drqn · 2025-11-01

**Soundness:** 3
**Presentation:** 3
**Contribution:** 3
**Rating:** 4
**Confidence:** 4

**Summary:**

The paper proposes OverRIDE, a history-aware decoding scheme to improve sample diversity without retraining the base LLM. The method keeps the original head to produce distribution, and adds a lightweight guide head (LoRA-style adapter on the output layer) that is updated online on previously generated samples to produce $q_t$. At each round, the next-token distribution used for sampling is reweighted by suppressing tokens that the guide head deems “common in prior rounds.” The paper provides an implementation aligned with parallel decoding (e.g., vLLM/SGLang-style KV reuse) and reports gains on code, math reasoning, and open-ended generation with small throughput drops that shrink for larger models.

**Strengths:**

- **Clear, pragmatic idea:** A *history-aware* decoding objective that can be layered onto existing stacks with minimal changes (only the output head; small adapter; online updates).
- **Engineering relevance:** Thoughtful integration with **KV-cache reuse** and round-synchronous updates; throughput overhead appears modest and decreases with model size.
- **Quality–diversity trade-off:** Reported increases in diversity (lower candidate similarity) with **PASS@k/MAUVE** not degrading and sometimes improving.
- **Simple hyperparameters:** Two main knobs—$\lambda$ (suppression strength) and adapter rank $r$—make the method operationally accessible.

**Weaknesses:**

1. **Temperature/top-p parity not rigorously controlled**
   The central claim is that OverRIDE provides more than generic flattening (i.e., increasing temperature). However, most comparisons are at author-chosen settings rather than **equal-diversity (equal entropy/KL)** conditions. Without controlled comparisons, it is hard to isolate the benefit beyond “being hotter.”

2. **Stability and guardrails of ratio reweighting**
   The method hinges on $p_t \propto (p/q_t)^\lambda p\$. If $q_t$ is small or miscalibrated on some tokens, the ratio can explode and induce overly aggressive exploration. The paper does not fully quantify **failure modes** (invalid/format-broken outputs) under large \(\lambda\), nor detail **safeguards** (logit clipping, $\epsilon$ -floors, temperature stacking).

3. **Guide-head “slight drift” is asserted but under-measured**
   The text claims the guide model “only drifts slightly,” but lacks systematic analysis of $\(\mathrm{KL}(q_t\|p)\)$ over rounds, **adapter-norm growth**, or how drift scales with $r$ and $\lambda$. This matters for safety and to understand when the guide becomes effectively an *anti-LM*.

4. **Parallel scheduling bias**
   The implementation updates head $t{+}1$ immediately after sampling round $t$ and enforces a **round-ordered schedule** for KV reuse. This could introduce **order bias** across candidates. There is no study of randomized/async updates or batch-size sensitivity to rule out engineering confounds.

5. **Ablations don’t fully decompose gains**
   Improvements in PASS@k could come from (a) genuine coverage of new solution modes vs. (b) more surface-different but semantically near misses. The paper lacks **semantic de-dup** analysis, **first-hit round distributions**, and **cost-normalized** comparisons (same steps/candidates).

**Questions:**

- **Equal-diversity controls:** Can you report results with *matched entropy* (or matched average NLL) against temperature/top-p, with significance testing?  .
- **Scheduling bias:** Does randomizing candidate update order or using async updates change results? Provide curves vs. batch size/KV hit rate.

---

> ### Author Response · Authors · 2025-11-23
> **Rebuttal from authors (1/2)**
>
> Thank you for your valuable comments. We hope the following responses can address your concerns.
>
> > **Q1. Temperature/top-p parity not rigorously controlled.**
>
> To compare OverRIDE with baseline methods at the same diversity level, we track the average negative log-likelihood (NLL) over all rounds of responses. Here we compare OverRIDE with baselines under different temperatures, so that baselines can cover different diversity conditions. We report the average average NLL over all rounds of responses:
>
> | Model | Method | PASS@10 (HumanEval) | NLL (HumanEval) | PASS@10 (MATH500) | NLL (MATH500) |
> |-------|-------|-------|-------|-------|-------|
> | Qwen-2.5-7B-Instruct | Baseline, $\tau=0.6$ | 81.4% | 17.3 | 86.8% | 37.2 |
> | | Baseline, $\tau=0.8$ | 83.0% | 20.3 | 87.1% | 43.8 |
> | | Baseline, $\tau=1.0$ | 85.5% | 25.2 | 87.2% | 53.6 |
> | | OverRIDE, $\tau=0.6$ | 86.4% | 20.0 | 87.3% | 41.3 |
> | Mistral-7B-Instruct-v0.3 | Baseline, $\tau=0.6$ | 54.9% | 57.6 | 36.4% | 80.8 |
> | | Baseline, $\tau=0.8$ | 56.1% | 69.5 | 36.2% | 98.8 |
> | | Baseline, $\tau=1.0$ | 58.1% | 89.3 | 36.2% | 126.9 |
> | | OverRIDE, $\tau=0.6$ | 56.9% | 64.5 | 37.4% | 86.4 |
>
> Comparing OverRIDE with the baseline at $\tau=0.8$ or even $\tau=1.0$, we can see that OverRIDE achieves better performance even with lower diversity (lower NLL). This validates the effectiveness of OverRIDE at the same diversity level.
>
> > **Q2. Stability and guardrails of ratio reweighting.**
>
> A safeguard is a good insight! Although we haven't experienced drift of guide model in our experiments on mainstream models (see **Q3**), a safeguard can protect the model from generating broken responses in rare extreme cases. Here we provide a safeguard to prevent the reweighting ratio from exploding. Specifically, users can choose to clip the reweighting ratio $(p(x) / q(x))^\lambda$ with a clipping threshold $\beta$:
>
> $$
> p_t(x) = \frac{1}{Z} ~~ clip \left( \left( \frac{p(x)}{q(x)} \right)^\lambda, ~\frac{1}{\beta}, ~\beta \right)  p(x),
> $$
>
> We have experimented with this safeguard on large $\lambda=1.5$, and show that our safeguard is effective in preventing the reweighting ratio from exploding, and helps achieve better decoding quality. Experiments are conducted on HumanEval using top-$p$ sampling with top-$p=0.9$ and $\tau=0.6$:
>
> | Model | $\beta$ | PASS@10 |
> |-------|-------|-------|
> | Qwen-2.5-7B-Instruct | No clip | 85.7% |
> | | 1.1 | 86.0% |
> | | 1.5 | **86.4%** |
> | | 2.0 | 86.1% |
> | Mistral-7B-Instruct-v0.3 | No clip | 36.0% |
> | | 1.1 | 34.2% |
> | | 1.5 | **39.0%** |
> | | 2.0 | **39.0%** |
>
> Note that this is only to provide an additional option for users to deal with extreme cases (e.g. when a much less diverse model is used). For moderate $\lambda$ values, the reweighting ratio is unlikely to explode, and the original tuning process is sufficient. Usually the optimal $\lambda$ falls into the normal situation.
>
> > **Q3. Guide-head “slight drift” is asserted but under-measured.**
>
> To validate our claim, we calculate the average KL divergence between the reweighted distribution $p_t(x)$ and the guide model distribution $q_t(x)$, $(KL(q_t | p))$, as you have mentioned. Experiments are conducted on MATH500 using top-$p$ sampling with top-$p=0.9$ and $\tau=0.6$:
>
> | Model | $KL(q_t \mid p)$  |
> |-------|-------|
> | Qwen-2.5-7B-Instruct | 0.26 |
> | Mistral-7B-Instruct-v0.3 | 0.07 |
>
> We can see that the KL divergence between the reweighted distribution and the guide model distribution is very small, indicating a slight drift. If the guide model drifts too much, we won't be able to achieve good decoding quality in our main experiments.

---

> ### Author Response · Authors · 2025-11-23
> **Rebuttal from authors (2/2)**
>
> > **Q4. OverRIDE introduces scheduling bias across candidates.**
>
> OverRIDE does **not** introduce scheduling bias. This is because the sampling process and the updating process are separated. There will not be cases where, some requests are sampled before updating reweighting heads, and others are sampled after.
>
> For further clarification, allow me to walk you through the process with vLLM as an example:
>
> When we ask vLLM to generate responses for some prompts, they are added as requests and duplicated in the case of parallel decoding. Based on the allocated memory for KV cache, the scheduler estimates how many requests can be processed as a continuous batch. These requests may be at different progress of decoding, and requests may be added (memory available) or removed (decoding finished) from the batch dynamically. Take an example in one of our Qwen-2.5-7B experiments, 19.77 GiB of memory is allocated for KV cache, and 370096 tokens are estimated to be processed by the model, thus the maximum concurrent requests is 45.18 for 8192 tokens per request.
>
> Once the batch is determined, the LLM engine will process all the requests in the batch concurrently. Hidden states are calculated, and then the next token is sampled for each request. Only after all the requests are sampled do we begin the fine-tuning process, and the corresponding adapters are updated based on which decoding round the request is at. Thus, there will not be cases where, some requests are sampled before updating reweighting heads, and others are sampled after, and causes scheduling bias.
>
> Yet, the batch size, as you have mentioned, may affect the performance of OverRIDE. This determines the granularity of how often the reweighting heads are updated. We conduct experiments with different amounts of GPU memory utilization, which determines how many requests can be processed as a continuous batch. The experiments are conducted on MATH500 using top-$p$ sampling with top-$p=0.9$ and $\tau=0.6$:
>
> | Model | GPU Memory Utilization | Max Concurrent Requests | PASS@10 (HumanEval) | PASS@10 (MATH500) |
> |-------|-------|-------|-------|-------|
> | Qwen-2.5-7B-Instruct | 0.4 (18GiB) | 3.38 | 86.5% | 87.2% |
> | | 0.6 (27GiB) | 21.46 | 86.6% | 87.2% |
> | | 0.8 (36GiB) | 45.18 | 86.4% | 87.3% |
> | Mistral-7B-Instruct-v0.3 | 0.4 (18GiB) | 3.38 | 57.2% | 37.2% |
> | | 0.6 (27GiB) | 21.46 | 56.7% | 37.7% |
> | | 0.8 (36GiB) | 45.18 | 56.9% | 37.4% |
>
> OverRIDE shows robustness on different batch sizes.
>
>
> > **Q5. Ablations don’t fully decompose gains. Improvements in PASS@k could come from (a) genuine coverage of new solution modes vs. (b) more surface-different but semantically near misses.**
>
> We argue that there is a **logical error** in your comment. More surface-different but semantically near misses will **not** contribute to the improvement of PASS@k. Indeed semantically similar responses may not improve PASS@k, but an increase in PASS@k must result from the model discovering new solution modes. Thus, the increase in PASS@k indicates that OverRIDE helps models find new solution modes. PASS@k is a more meaningful and direct metric to validate the effectiveness of OverRIDE than semantic analysis.
>
> Cost-normalized comparison is **exactly** what we have conducted in our main experiments. We control the computational cost by setting the allowed decoding rounds (candidates) as 5 or 10, and then compare the accuracy, resulting in metrics PASS@5 and PASS@10.
>
> Yet, first-hit round analysis is an interesting idea. This validates that OverRIDE can effectively reduce the computational cost for finding a new solution. Below are the average first-hit round results on HumanEval and MATH500 using top-$p$ sampling with top-$p=0.9$ and $\tau=0.6$:
>
> | Model | Method | HumanEval First-hit Round | MATH500 First-hit Round |
> |-------|-------|-------|-------|
> | Qwen-2.5-7B-Instruct | Baseline | 1.72 | 1.45 |
> | | OverRIDE | **1.67** | **1.44** |
> | Mistral-7B-Instruct-v0.3 | Baseline | 2.57 | 3.60 |
> | | OverRIDE | **2.49** | **3.52** |

---

> ### Author Response · Authors · 2025-11-26
> **Thank you for your efforts in improving OverRIDE!**
>
> Dear reviewer Drqn,
>
> Thanks to your suggestions on extensive experiments, we are able to further validate that OverRIDE is indeed an effective and robust decoding method.
>
> If you have further questions and doubts, we are always glad to provide more clarification.
>
> Otherwise, if you find our work more solid after we respond, could you consider raising your score to the corresponding level?

---

### Official Review · Reviewer_wTcW · 2025-11-01

**Soundness:** 2
**Presentation:** 3
**Contribution:** 3
**Rating:** 4
**Confidence:** 5

**Summary:**

The paper introduces OverRIDE, an iterative reweighting decoding method designed to increase output diversity while preserving task performance. OverRIDE introduces test-time trainable output-head adapters that iteratively reweight next-token logits to suppress tokens leading to previously seen patterns, steering decoding away from already-explored distributions. The adapters are confined to the output head, updated after each round, and synchronized with sampling, which makes the method compatible with parallel inference in serving systems. Combined with standard sampling, this history-aware reweighting increases diversity while preserving quality across code, math, and open-ended generation tasks.

**Strengths:**

1. The experimental coverage is broad, including two LLM families (Qwen2.5 and Mistral), five parameter scales from 3B to 72B, and four standard benchmarks.
2. OverRIDE shows consistent diversity improvements over common decoding baselines in nearly all settings, with pass@k accuracy gains, while incurring only a modest slowdown.

**Weaknesses:**

1. For the CCNews open-ended long-form generation in Table 2, Distinct-n and Self-BLEU may be more appropriate diversity or similarity metrics than paragraph-level embedding cosine similarity.
2. Comparisons focus on basic sampling schemes and min-p. How does your method perform compared with other classic baselines such as beam search and group beam search?
3. Training and implementation details are under-specified. The core Output head adapters lack information on parameter counts, learning rates, and other settings. Section 3.4 reports only throughput in tokens per second without the hardware configuration.
4. When restricted to a single iteration, OverRIDE has no effect and collapses to the baseline, which limits scenarios that demand single-pass decoding.

**Questions:**

1. There is a discrepancy between your Table 1 and the Qwen-2.5 technical report: on HumanEval, the official PASS@1 is 52.9 for the 7B base model (Table 4 [1]) and 84.8 for the 7B Instruct model (Table 8 [1]), whereas your Table 1 lists 64.9 under “greedy.” After reviewing your released code, I see you evaluate Qwen-2.5-X-B-Instruct rather than the base Qwen-2.5-X-B. Please verify why your evaluation diverges from the official numbers, and standardize the model naming in the paper (or re-run the baselines with the base model) to avoid confusion.
2. Additionally, I would like to know the exact evaluation setup for CCNews reported in Table 2. The relevant code does not appear to be included in your supplementary materials.
3. In Table 3, row 315 shows Qwen-2.5-14B with a notably lower HumanEval PASS@1  (greedy)  than other sizes of the same family. Is this a typographical error?


[1] Qwen Team. (2025). *Qwen2.5 Technical Report*. arXiv. arXiv:2412.15115.

---

> ### Author Response · Authors · 2025-11-23
> **Rebuttal from authors (1/3)**
>
> Thank you for your valuable comments. We hope the following responses can address your concerns.
>
> > **Q1. Self-BLEU and Distinct-n on CCNews.**
>
> Below we provide the results on CCNews with Self-BLEU [1] and Distinct-n [2]. Experiments are conducted using top-$p$ sampling with top-$p=0.9$ and $\tau=0.6$. The implementation of these two metrics are as follows:
>
> - Self-BLEU: We use the Github implementation of SacreBLEU [3] to obtain BLEU score. SacreBLEU provides a standard tokenization scheme that avoids ambiguity in BLEU score calculation caused by different parsing methods. Self-BLEU is calculated by averaging the pairwise BLEU scores between responses, which is the same as the original paper. Lower self-BLEU indicates higher diversity.
>
> - Distinct-n: We reuse the tokenizer in SacreBLEU [3] to tokenize the generated text, and then calculate Distinct-1 and Distinct-2 scores using the implementation in PaddlePaddle. We take the average Distinct-n score of all the 10 rounds of responses. Higher Distinct-n indicates higher diversity.
>
> | Model | Method | Self-BLEU ($\downarrow$) | Distinct-1 ($\uparrow$) | Distinct-2 ($\uparrow$) |
> |-------|-------|-------|-------|-------|
> | Qwen-2.5-7B-Instruct | Baseline | 11.02 | 0.467 | **0.774** |
> | | OverRIDE | **5.32** | **0.480** | 0.770 |
> | Mistral-7B-Instruct-v0.3 | Baseline | 13.65 | 0.545 | 0.850 |
> | | OverRIDE | **6.75** | **0.567** | **0.862** |
>
> > **Q2. Comparison with other classic baselines.**
>
> Besides Beam Search, we also compare with two additional baselines, SimCTG [4] and Contrastive Decoding [5]. Here are the settings for these baselines:
>
> - Beam Search: We use the implementation integrated in vLLM. We set beam size $beam=30$.
>
> - SimCTG: We use the implementation integrated in Huggingface transformers. We set degeneration penalty $\alpha=0.6$, and top-$k$ prediction $k=4$.
>
> - Contrastive Decoding: We use the implementation from their Github repo. Since they only support models before 2023, and haven't responded to requests for adding recent models, we are only able to compare with OverRIDE on GPT2-XL. Following the original implementation, we set the amateur model as GPT2, amateur model temperature $\tau_{amateur}=0.5$,and plausibility threshold $\alpha=0.1$. For OverRIDE, we set $\lambda=0.4$.
>
> Beam Search & SimCTG:
>
> | Model | Method | PASS@10 (HumanEval) | CodeBLEU (HumanEval) | PASS@10 (MATH500) | Similarity (MATH500) | MAUVE (CCNews) | Similarity (CCNews) |
> |-------|-------|-------|-------|-------|-------|-------|-------|
> | Qwen-2.5-7B-Instruct| Beam Search | 73.2% | 0.886 | 77.2% | 0.994 | 0.727 | 0.990 |
> | | SimCTG | 77.2% | 0.747 | 86.8% | 0.954 | **0.976** | 0.756 |
> | | OverRIDE | **86.4%** | **0.610** | **87.3%** | **0.936** | 0.938 | **0.678** |
> | Mistral-7B-Instruct-v0.3 | Beam Search | 41.5% | 0.916 | 19.4% | 0.986 | 0.799 | 0.993 |
> | | SimCTG | 56.1% | 0.528 | 36.8% | 0.868 | 0.915 | 0.758 |
> | | OverRIDE | **56.9%** | **0.474** | **37.4%** | **0.862** | **0.952** | **0.703** |
>
> Contrastive Decoding:
>
> | Model | Method | MAUVE (CCNews) | Similarity (CCNews) |
> |-------|-------|-------|-------|
> | GPT2-XL | CD | 0.752 | 0.676 |
> | | OverRIDE | **0.846** | **0.664** |
>
> [1] Zhu Y, Lu S, Zheng L, et al. Texygen: A benchmarking platform for text generation models[C]//The 41st international ACM SIGIR conference on research & development in information retrieval. 2018: 1097-1100.
>
> [3] Li J, Galley M, Brockett C, et al. A diversity-promoting objective function for neural conversation models[C]//Proceedings of the 2016 conference of the North American chapter of the association for computational linguistics: human language technologies. 2016: 110-119.
>
> [2] Post M. A call for clarity in reporting BLEU scores[J]. arXiv preprint arXiv:1804.08771, 2018.
>
> [4] Su Y, Lan T, Wang Y, et al. A contrastive framework for neural text generation[J]. Advances in Neural Information Processing Systems, 2022, 35: 21548-21561.
>
> [5] Li X L, Holtzman A, Fried D, et al. Contrastive decoding: Open-ended text generation as optimization[C]//Proceedings of the 61st annual meeting of the association for computational linguistics (volume 1: Long papers). 2023: 12286-12312.

---

> ### Author Response · Authors · 2025-11-23
> **Rebuttal from authors (2/3)**
>
> > **Q3. Training and implementation details.**
>
> In terms of the adapter for the output head, matrix $W_A$ is initialized with normal distribution $N(0, 0.02)$, and matrix $W_B$ is initialized with zeros, which is the same as the default settings in the PEFT library. The learning rate is set to $1e-3$.
>
> We conduct all experiments on 8 NVIDIA L40S GPUs. We use the default settings of vLLM except for the following: gpu memory utilization is set to 0.8 for all the experiments; we use a tensor parallel size of 1 for 3B and 7B models, 2 for 14B models, 4 for 32B models, and 8 for 72B models.
>
> Most of these settings have been specified in Appendix B, but thank you for your attention to the implementation details. We appreciate your emphasis on reproducibility, which is crucial for the research community.
>
> In case you still have concerns on the runtime overhead of OverRIDE in practice, we further provide a memory cost analysis. Memory cost is calculated by averaging a 10ms-interval snapshot with `MemorySnapshot` in the naive vLLM utils file. Experiments are conducted on MATH500 dataset, using top-$p$ sampling with top-$p=0.9$ and $\tau=0.6$, which is the same as the throughput experiment in our paper.
>
> | Model | Model Size (GiB) | OverRIDE Runtime Memory (GiB) | $\Delta$ over model size (%) |
> |-------|-------|-------|-------|
> | Qwen-2.5-7B-Instruct | 14.19 | 0.7500 | 5.3% |
> | Qwen-2.5-14B-Instruct | 27.51 | 1.1997 | 4.4% |
> | Qwen-2.5-32B-Instruct | 61.03 | 2.6370 | 4.3% |
> | Qwen-2.5-72B-Instruct | 135.43 | 6.3874 | 4.7% |
>
> This shows that OverRIDE introduces minimal memory cost. The runtime OverRIDE memory cost is calculated by the difference between vLLM runtime memory with and without OverRIDE. Since vLLM pre-allocates extra GPU memory for KV cache (runtime memory takes up a fixed proportion), for a clear comparison, we show the memory cost against the original model size.

---

> ### Author Response · Authors · 2025-11-23
> **Rebuttal from authors (3/3)**
>
> > **Q4. When generating with a single iteration, OverRIDE has no effect and collapses to the baseline.**
>
> The goal of OverRIDE is to generate multiple diverse responses. We argue that this is a strong motivation with plentiful scenarios, for example:
> - Proposing multiple candidates for users in open-generation queries;
> - Diverse rollouts for exploration in RL training;
> - Consensus methods for enhancing reasoning capabilities by voting on multiple responses [6] [7].
>
> > **Q5. Discrepancy between Table 1 and Qwen-2.5 technical report. Qwen-2.5-14B has a notably lower HumanEval PASS@1.**
>
> Thank you for the reminder. We use `Qwen-2.5-XB-Instruct` models for all the experiments in our paper. We have updated the corresponding descriptions in our paper. We have tried multiple prompting methods, but could not reach the performance claimed in the official report. To avoid introducing bias by our implementation, we adopt the same prompting strategy as EvalPlus [9]. Furthermore, we believe our comparison is still fair, since the same setting is used for all the experiments.
>
> For `Qwen-2.5-14B-Instruct`, it is surprising that the performance on HumanEval is lower than smaller models, but this is indeed the case. There is actually a typo that the PASS@10 result of the baseline is lower, being 71.1 instead of 77.1. We have corrected this typo in our paper. Our experiments are averaged over multiple independent runs, and the prompting strategy is the same. We can provide you with the HumanEval results with $\pm 1\sigma$:
>
> | Model | Method | PASS@5 | PASS@10 |
> |-------|-------|-------|-------|
> | Qwen-2.5-14B-Instruct | Baseline | 63.4$\pm$1.7 | 71.1$\pm$1.5 |
> | | OverRIDE | **70.5$\pm$2.6** | **77.2$\pm$1.3** |
>
> If our explanation still doesn't convince you, we have a better proposal: to experiment on a more challenging coding benchmark **LiveCodeBench** [8]. We directly use the official Github implementation of LiveCodeBench with release-v6, so that any bias introduced by our implementation is not possible. Since they don't support Mistral models, we report additional results on `Llama-3-8B-Instruct`, which also further validates the effectiveness of OverRIDE on other models and difficult tasks.
>
> Since LiveCodeBench supports using vLLM for fast inference, we directly transfer our implementation of OverRIDE to their benchmark. This further shows how convenient it is to integrate OverRIDE with existing inference frameworks. Experiments are conducted using top-$p$ sampling with top-$p=0.9$ and $\tau=0.6$:
>
> | Model | Method | PASS@5 | PASS@10 | CodeBLEU |
> |-------|-------|-------|-------|-------|
> | Qwen-2.5-7B-Instruct | Baseline | 33.2% | 36.8% | 0.515 |
> | | OverRIDE | **35.2%** | **39.8%** | **0.433** |
> | Llama-3-8B-Instruct | Baseline | 19.8% | 22.7% | 0.491 |
> | | OverRIDE | **20.8%** | **24.4%** | **0.440** |
> | Qwen-2.5-72B-Instruct | Baseline | 49.4% | 51.5% | 0.645 |
> | | OverRIDE | **50.6%** | **53.2%** | **0.588** |
>
>
> > **Q6. Evaluation setup for CCNews.**
>
> We use `mauve` library from the official Github repo of MAUVE [10] to calculate MAUVE score. Specifically, the code is shown as follows:
>
> ```
> import mauve
>
> out = mauve.compute_mauve(
>     p_text=ref_list,
>     q_text=pred_list,
>     max_text_length=256,
>     featurize_model_name='gpt2'
> )
> mauve_score = out.mauve
> ```
>
> Here, `ref_list` is the list of all the human written news samples, and `pred_list` is the list of all 10 rounds of the model responses to all the news sample contexts. Since the library only supports GPT-2 or BERT as the feature extraction model, we use GPT-2, which is the same as the official Github implementation of Contrastive Decoding [5].
>
> ---
>
> [5] Li X L, Holtzman A, Fried D, et al. Contrastive decoding: Open-ended text generation as optimization[C]//Proceedings of the 61st annual meeting of the association for computational linguistics (volume 1: Long papers). 2023: 12286-12312.
>
> [6] Wang X, Wei J, Schuurmans D, et al. Self-consistency improves chain of thought reasoning in language models[J]. arXiv preprint arXiv:2203.11171, 2022.
>
> [7] Yao S, Yu D, Zhao J, et al. Tree of thoughts: Deliberate problem solving with large language models[J]. Advances in neural information processing systems, 2023, 36: 11809-11822.
>
> [8] Jain N, Han K, Gu A, et al. Livecodebench: Holistic and contamination free evaluation of large language models for code[J]. arXiv preprint arXiv:2403.07974, 2024.
>
> [9] Liu J, Xia C S, Wang Y, et al. Is your code generated by chatgpt really correct? rigorous evaluation of large language models for code generation[J]. Advances in Neural Information Processing Systems, 2023, 36: 21558-21572.
>
> [10] Pillutla K, Swayamdipta S, Zellers R, et al. Mauve: Measuring the gap between neural text and human text using divergence frontiers[J]. Advances in Neural Information Processing Systems, 2021, 34: 4816-4828.

---

> ### Author Response · Authors · 2025-11-26
> **Thank you for your efforts in improving OverRIDE!**
>
> Dear reviewer wTcW,
>
> Thanks to your suggestions on extensive experiments, we are able to further validate that OverRIDE is indeed an effective and robust decoding method.
>
> If you have further questions and doubts, we are always glad to provide more clarification.
>
> Otherwise, if you find our work more solid after we respond, could you consider raising your score to the corresponding level?

---

### Official Review · Reviewer_om5c · 2025-11-11

**Soundness:** 2
**Presentation:** 3
**Contribution:** 2
**Rating:** 4
**Confidence:** 4

**Summary:**

This paper proposes a novel decoding method called OverRIDE, increasing semantic diversity in the generation of text by LLMs. It dynamically adjusts the probability distribution of the next word in the inference stage, encouraging the model to explore new and low-probability generation paths. The experiment proved that this method can enhance semantic diversity and achieve better results in multiple sampling processes.

**Strengths:**

1. The homogenization of the content generated by LLMs is a widespread problem that needs to be addressed.
2. The authors designed and implemented a parallel algorithm based on high-performance inference frameworks, which is beneficial for the use of the open-source community.

**Weaknesses:**

My primary concerns relate to the comprehensiveness of the experimental evaluation. To be confident in the paper's claims, the following points need to be addressed:
1.  Incomplete Efficiency Evaluation. The paper claims "minimal loss of efficiency" based solely on throughput. For a method that introduces training at inference time, this is insufficient. A more thorough analysis of computational overhead is critical. Please report Time To First Token metric for user-perceived latency, and the overall memory cost compared with normal sampling.
2.  Limited and Potentially Dated Datasets. The experiments are conducted on relatively simple tasks. Several studies have demonstrated the crucial role of the "critical token" in ensuring the correctness of the model's responses. Unselectively suppressing the tokens generated in the first round may affect the model's accuracy. The benchmarks evaluated by the authors, such as MATH-500 and GSM8k, are not challenging for Qwen2.5 and might overlook this issue. It is recommended to add more cutting-edge data, at least including AIME.
3.  Missing Comparison with key baselines. The paper compares OverRIDE with popular sampling techniques but omits a comparison with other advanced decoding methods that also aim to improve generation quality and diversity. A direct comparison with Contrastive Search is essential. Both methods modify the decoding objective to balance model likelihood with a penalty term. A head-to-head comparison on both quality and diversity metrics is necessary to properly contextualize the contributions of OverRIDE.
4.  Concern of Hyperparameter Sensitivity Analysis. The proposed algorithm likely depends on a key hyperparameter $\lambda$. Although the author conducted the sensitivity experiment, \lambda may still be a challenging parameter. Users will not have fine-grained prior knowledge during the usage process. Robustness and usability of the method are supposed to be verified through a broader model family and more difficult data (as mentioned in the second concern)

**Questions:**

1. Can you add evaluations on the time to first token metric and the overall memory cost compared with normal sampling?
2. Does OverRIDE also have considerable superiority on more challenging benchmarks? Can robustness research be conducted on more difficult data such as AIME and LiveCodeBench?
3. The recent contrastive decoding method mentioned by the author should be compared with OverRIDE [1].
4. Can you provide fine-grained sensitivity analysis on the lambda performance of more model families on more challenging benchmarks?

[1] Su, Yixuan, et al. "A contrastive framework for neural text generation." Advances in Neural Information Processing Systems 35 (2022): 21548-21561.

---

> ### Author Response · Authors · 2025-11-23
> **Rebuttal from authors (1/3)**
>
> Thank you for your valuable comments. We hope the following responses can address your concerns.
>
> > **Q1. Evaluation on first token metric and memory cost.**
>
> Below we provide the average time to first token (TTFT) and the additional memory cost of OverRIDE. TTFT is calculated using the native metric from vLLM: `ttft = output.metrics.first_token_ts - output.metrics.scheduled_ts`. Memory cost is calculated by averaging a 10ms-interval snapshot with `MemorySnapshot` in the naive vLLM utils file. Experiments are conducted on MATH500 dataset, using top-$p$ sampling with top-$p=0.9$ and $\tau=0.6$, which is the same as the throughput experiment in our paper.
>
> TTFT:
>
> | Model | Baseline TTFT (ms) | OverRIDE TTFT (ms) |
> |-------|-------|-------|
> | Qwen-2.5-7B-Instruct | 136.0 | 158.8 |
> | Qwen-2.5-14B-Instruct | 164.8 | 197.3 |
> | Qwen-2.5-32B-Instruct | 268.3 | 301.5 |
> | Qwen-2.5-72B-Instruct | 574.2 | 639.1 |
>
> An important feature to notice is that, since OverRIDE doesn't adjust the first round of generation, users can always receive the initial response within the **original latency**. For susequent rounds, TTFT still stays at a user-friendly level. From the user's perspective, such a behavior is highly acceptable. In scenarios that don't involve user interaction (e.g., parallel rollouts for RL training), TTFT becomes a less critical metric, and our thoughput still stays competitive.
>
> Memory:
>
> | Model | Model Size (GiB) | OverRIDE Runtime Memory (GiB) | $\Delta$ over model size (%) |
> |-------|-------|-------|-------|
> | Qwen-2.5-7B-Instruct | 14.19 | 0.7500 | 5.3% |
> | Qwen-2.5-14B-Instruct | 27.51 | 1.1997 | 4.4% |
> | Qwen-2.5-32B-Instruct | 61.03 | 2.6370 | 4.3% |
> | Qwen-2.5-72B-Instruct | 135.43 | 6.3874 | 4.7% |
>
> This shows that OverRIDE introduces minimal memory cost. The runtime OverRIDE memory cost is calculated by the difference between vLLM runtime memory with and without OverRIDE. Since vLLM pre-allocates extra GPU memory for KV cache (runtime memory takes up a fixed proportion), for a clear comparison, we show the memory cost against the original model size.
>
> > **Q2. Evaluation on more challenging benchmarks.**
>
> Below we provide the results on AIME2024 and LiveCodeBench. For AIME2024, since it only includes 30 questions, we average the results on >10 independent runs. Additionally, since `Qwen2.5-7B-Instruct` and `Mistral-7B-Instruct-v0.3` fail on all questions, we report results on `Qwen2.5-72B-Instruct`, `DeepSeek-R1-Distill-Qwen-7B`, and `DeepSeek-R1-Distill-Llama-8B`. This further validates the effectiveness of OverRIDE on a broader range of models.
>
> For LiveCodeBench, we use the original benchmark in their Github repo with release-v6. Since they don't support Mistral models, we report addtional results on `Llama-3-8B-Instruct`. Since LiveCodeBench supports using vLLM for fast inference, we directly transfer our implementation of OverRIDE to their benchmark. This further shows how convenient it is to integrate OverRIDE with existing inference frameworks.
>
> Experiments are conducted using top-$p$ sampling with top-$p=0.9$ and $\tau=0.6$.
>
> AIME2024:
>
> | Model | Method | PASS@5 | PASS@10 | Similarity |
> |-------|-------|-------|-------|-------|
> | DeepSeek-R1-Distill-Qwen-7B | Baseline | 53.4% | 55.4% | 0.865 |
> | | OverRIDE | **54.2%** | **56.5%** | **0.832** |
> | DeepSeek-R1-Distill-Llama-8B | Baseline | 45.2% | 51.0% | 0.855 |
> | | OverRIDE | **46.7%** | **53.3%** | **0.803** |
> | Qwen2.5-72B-Instruct | Baseline | 16.0% | 20.7% | 0.926 |
> | | OverRIDE | **16.9%** | **22.9%** | **0.847** |
>
> LiveCodeBench:
>
> | Model | Method | PASS@5 | PASS@10 | CodeBLEU |
> |-------|-------|-------|-------|-------|
> | Qwen-2.5-7B-Instruct | Baseline | 33.2% | 36.8% | 0.515 |
> | | OverRIDE | **35.2%** | **39.8%** | **0.433** |
> | Llama-3-8B-Instruct | Baseline | 19.8% | 22.7% | 0.491 |
> | | OverRIDE | **20.8%** | **24.4%** | **0.440** |
> | Qwen-2.5-72B-Instruct | Baseline | 49.4% | 51.5% | 0.645 |
> | | OverRIDE | **50.6%** | **53.2%** | **0.588** |

---

> ### Author Response · Authors · 2025-11-23
> **Rebuttal from authors (2/3)**
>
> > **Q3. Comparison with other advanced decoding methods.**
>
> Besides Contrastive Decoding [1], we also compare with two additional baselines, SimCTG [2] and Beam Search. Here are the settings for these baselines:
>
> - Contrastive Decoding (CD): We use the implementation from their Github repo. Since they only support models before 2023, and haven't responded to requests for adding recent models, we are only able to compare with OverRIDE on `GPT2-XL`. Following the original implementation, we set the amateur model as `GPT2`, amateur model temperature $\tau_{amateur}=0.5$,and plausibility threshold $\alpha=0.1$. For OverRIDE, we set $\lambda=0.4$.
>
> - SimCTG: We use the implementation integrated in Huggingface transformers. We set degeneration penalty $\alpha=0.6$, and top-$k$ prediction $k=4$.
>
> - Beam Search: We use the implementation integrated in vLLM. We set beam size $beam=30$.
>
> Contrastive Decoding:
>
> | Model | Method | MAUVE (CCNews) | Similarity (CCNews) |
> |-------|-------|-------|-------|
> | GPT2-XL | CD | 0.752 | 0.676 |
> | | OverRIDE | **0.846** | **0.664** |
>
> SimCTG & Beam Search:
>
> | Model | Method | PASS@10 (HumanEval) | CodeBLEU (HumanEval) | PASS@10 (MATH500) | Similarity (MATH500) | MAUVE (CCNews) | Similarity (CCNews) |
> |-------|-------|-------|-------|-------|-------|-------|-------|
> | Qwen-2.5-7B-Instruct| Beam Search | 73.2% | 0.886 | 77.2% | 0.994 | 0.727 | 0.990 |
> | | SimCTG | 77.2% | 0.747 | 86.8% | 0.954 | **0.976** | 0.756 |
> | | OverRIDE | **86.4%** | **0.610** | **87.3%** | **0.936** | 0.938 | **0.678** |
> | Mistral-7B-Instruct-v0.3 | Beam Search | 41.5% | 0.916 | 19.4% | 0.986 | 0.799 | 0.993 |
> | | SimCTG | 56.1% | 0.528 | 36.8% | 0.868 | 0.915 | 0.758 |
> | | OverRIDE | **56.9%** | **0.474** | **37.4%** | **0.862** | **0.952** | **0.703** |
>
> [1] Li X L, Holtzman A, Fried D, et al. Contrastive decoding: Open-ended text generation as optimization[C]//Proceedings of the 61st annual meeting of the association for computational linguistics (volume 1: Long papers). 2023: 12286-12312.
>
> [2] Su Y, Lan T, Wang Y, et al. A contrastive framework for neural text generation[J]. Advances in Neural Information Processing Systems, 2022, 35: 21548-21561.

---

> ### Author Response · Authors · 2025-11-23
> **Rebuttal from authors (3/3)**
>
> > **Q4. Fine-grained sensitivity analysis**
>
> Besides MATH500 that has been shown in our paper, we further conduct sensitivity analysis on HumanEval and LiveCodeBench. We suggest not to analyze AIME2024 since results are of high variance and may not be representative for sensitivity analysis. Furthermore, we add `Llama-3-8B-Instruct` as a new model. Below we report the PASS@10 results of OverRIDE under different $\lambda$ values.
>
> HumanEval:
>
> | Model | $\lambda=0.2$ | $\lambda=0.4$ | $\lambda=0.6$ | $\lambda=0.8$ | $\lambda=1.0$ |
> |-------|-------|-------|-------|-------|-------|
> | Qwen-2.5-7B-Instruct | 83.5% | 85.4% | 85.8% | **86.4%** | 86.3% |
> | Mistral-7B-Instruct-v0.3 | 56.4% | **56.9%** | 55.1% | 54.9% | 54.3% |
> | Llama-3-8B-Instruct | 72.0% | **73.8%** | 73.5% | 73.5% | 73.2% |
>
> LiveCodeBench:
>
> | Model | $\lambda=0.2$ | $\lambda=0.4$ | $\lambda=0.6$ | $\lambda=0.8$ | $\lambda=1.0$ |
> |-------|-------|-------|-------|-------|-------|
> | Qwen-2.5-7B-Instruct | 37.5% | 37.0% | 38.3% | **39.8%** | 38.2% |
> | Llama-3-8B-Instruct | 23.1% | **24.4%** | 24.3% | 24.2% | 24.1% |
>
> The hyperparameter $\lambda$ is actually extremely easy and efficient to tune. Since $\lambda$ controls the intensity of the reweighting ratio $(p(x) / q(x))^\lambda$, when setting to $\lambda=0$, OverRIDE is the same as the original model's next-token distribution. Thus, we recommend to tune $\lambda$ starting from 0 and gradually increasing it until the desired diversity is achieved. For less diverse models like Qwen, we have shown in our experiment that the optimal $\lambda$ is around 0.8. For more diverse models like Mistral and Llama, the optimal $\lambda$ is even lower, thus relieves the effort of hyperparameter tuning.
>
> The best practice is to tune $\lambda$ on a small validation set. When using OverRIDE on a new model or task, we recommend to take around 100 samples and tune $\lambda$ starting from 0. Although the hardware configuration / memory settings / model / task may vary, with the help of our parallel implementation on LLM serving systems like vLLM, tuning one round typically takes **minutes**. Furthermore, reaching the optimal $\lambda$ usually takes only a few rounds.
>
> In case you still have concerns about the tuning process when $\lambda$ is too high, we additionally provide a safeguard to prevent the reweighting ratio from exploding. Specifically, users can choose to clip the reweighting ratio $(p(x) / q(x))^\lambda$ with a clipping threshold $\beta$:
>
> $$
> p_t(x) = \frac{1}{Z} ~~ clip \left( \left( \frac{p(x)}{q(x)} \right)^\lambda, ~\frac{1}{\beta}, ~\beta \right)  p(x),
> $$
>
> We have experimented with this safeguard on large $\lambda=1.5$, and show that our safeguard is effective in preventing the reweighting ratio from exploding. Experiments are conducted on HumanEval using top-$p$ sampling with top-$p=0.9$ and $\tau=0.6$:
>
> | Model | $\beta$ | PASS@10 |
> |-------|-------|-------|
> | Qwen-2.5-7B-Instruct | No clip | 85.7% |
> | | 1.1 | 86.0% |
> | | 1.5 | **86.4%** |
> | | 2.0 | 86.1% |
> | Mistral-7B-Instruct-v0.3 | No clip | 36.0% |
> | | 1.1 | 34.2% |
> | | 1.5 | **39.0%** |
> | | 2.0 | **39.0%** |
>
> Note that this is only to provide an additional option for users to tune on extreme cases (e.g. when a much less diverse model is used). For moderate $\lambda$ values, the reweighting ratio is unlikely to explode, and the original tuning process is sufficient. Usually the optimal $\lambda$ falls into the normal situation.

---

> ### Author Response · Authors · 2025-11-26
> **Thank you for your efforts in improving OverRIDE!**
>
> Dear reviewer om5c,
>
> Thanks to your suggestions on extensive experiments, we are able to further validate that OverRIDE is indeed an effective and robust decoding method.
>
> If you have further questions and doubts, we are always glad to provide more clarification.
>
> Otherwise, if you find our work more solid after we respond, could you consider raising your score to the corresponding level?

---

### Comment · Area_Chair_my87 · 2025-11-28

Dear Reviewers,

The authors have responded to your reviews. Please engage in the discussion and evaluate the authors’ rebuttal to check whether your comments have been adequately addressed, and determine whether you would like to adjust your evaluations.

Best,

Your AC

---

### Author Response · Authors · 2025-11-29
**Review & Rebuttal Summary**

Overall, all the reviewers agree with our motivation and methodology, raising no concerns. The main concerns are about the experimental evaluation to further validate our claims. We have addressed all the concerns with extensive experiments, and validate that OverRIDE is an effective and robust decoding method for enhancing diversity.

---

### Main Strengths

- **Stong motivation:** The homogenization and diversity of LLM responses are a widespread problem. *(Reviewer om5c, BHNo)*

- **Elegant and pragmatic methodology:** OverRIDE directly learns from history generations, and introduces minimal changes. *(Reviewer Drqn, BHNo)*

- **Strong experimental results:** OverRIDE shows consistent improvements on 4 benchmarks (HumanEval, MATH500, GSM8K, CCNews), 2 model families (Qwen, Mistral), and 5 model scales (3B to 72B). We add 2 additional model families (Llama, DeepSeek), and 2 additional benchmarks (AIME2024, LiveCodeBench) in our rebuttal. *(Reviewer wTcW, BHNo)*

- **Real-world deployment:** We provide a parallel implementation on vLLM with minimal slowdown, which is beneficial for the use of the open-source community. We have prepared the code on 2 vLLM versions (v0.8.5.post1 and v0.11.1), and 2 advanced benchmarks (LiveCodeBench, EvalPlus). We will release the code upon acceptance. *(Reviewer om5c, Drqn, BHNo)*

---

### Main Weaknesses and Rebuttals

- **Comparison with other decoding methods:** We compare OverRIDE with 3 additional baselines: Contrastive Decoding, SimCTG, and Beam Search. We show that OverRIDE outperforms these baselines on various models and benchmarks. *(Reviewer om5c, wTcW, BHNo)*

- **Hyperparameter sensitivity analysis:** We conduct an additional sensitivity analysis of the hyperparameter $\lambda$ on 2 benchmarks (HumanEval, LiveCodeBench) and 3 model families (Qwen, Mistral, Llama). We provide a simple tutorial for users to tune $\lambda$ easily within **minutes**. Additionally, inspired by reviewer Drqn, we add a safeguard to prevent failures on extreme cases with high $\lambda$ values, and validate that the safeguard is effective. *(Reviewer om5c, Drqn, BHNo)*

- **Necessity of our reweighting mechanism:** We clarify the necessity and simplicity of the proposed reweighting formula based on our motivation. Additionally, we provide a theoretical explanation for why OverRIDE balances exploration and exploitation. *(Reviewer BHNo)*

- **Evaluation on more challenging benchmarks:** We provide additional results on 2 challenging benchmarks (AIME2024, LiveCodeBench), and show that OverRIDE still achieves consistent improvements. *(Reviewer om5c)*

- **Runtime cost analysis:** Besides throughput analysis in our paper, we evaluate the average time to first token (TTFT) and the additional memory cost of OverRIDE. These additional runtime metrics show that OverRIDE preserves a smooth user experience, and introduces minimal memory cost. *(Reviewer om5c)*

- **Implementation details for reproducibility:** We reiterate the detailed implementation of OverRIDE from Appendix B. We also show the implementation details of the prompting strategies and MAUVE metrics for reproducibility. *(Reviewer wTcW)*

---

### Potentially Low-quality Reviews

Reviewer Drqn proposed many novel concepts without reference (e.g. temperature stacking, semantic de-dup). Their comments also contain logical errors (please see Q5). Nevertheless, we have carefully addressed all the concerns with extensive experiments.

---

### Meta-Review · Area_Chair_2n9f · 2026-01-04

**Summary:**

## Summary

This paper proposes a novel decoding method called OverRIDE, increasing semantic diversity in the generation of text by LLMs. It dynamically adjusts the probability distribution of the next word in the inference stage, encouraging the model to explore new and low-probability generation paths.  To validate the method, the paper conducts experiments based on  two LLM families (Qwen2.5 and Mistral), five parameter scales from 3B to 72B, and four standard benchmarks.

## Overall Score

om5c: 4 (no response)

wTcW:  4 (no response)

Drqn: 4 (no response)

BHNo: 6 (no response)

## Concerns

* Incomplete Efficiency Evaluation (om5c, wTcW, Drqn)
* Limited and Potentially Dated Datasets (om5c)
* Missing some baselines, like other advanced decoding methods (om5c, BHNo)
* More detail about the implementation (om5c,wTcW, BHNo)
* Limitation in single-pass decoding (wTcW)
* Temperature/top-p parity not rigorously controlled (Drqn)
* Limited Ablation design (Drqn)
* No theoretical guarantees (BHNo)

## Conclusion:

Overall, during the review phase, most of the reviewers (om5c, wTcW, and Drqn) weren't fully satisfied with the experimental design. They raised concerns about the choice of metrics, baselines, and some implementation details. Reviewer BHNo, on the other hand, leaned toward acceptance. During the discussion, the authors did a solid job addressing the reviewers' questions and concerns, but none of the reviewers followed up afterward.

This paper has potential to be stronger with some improvements to the experimental design and theoretical justification.

**Reviewer Concerns:**

Refer to Summary

**Reviewer Scores:**

Refer to Summary

---

### Decision · Program_Chairs · 2026-01-26

Accept (Poster)